# Diffusion models are optimal for hypothesis testing

## Abstract

Diffusion models have demonstrated powerful generative capabilities, but their potential in statistical hypothesis testing remains underexplored. The score-based paradigm of diffusion formulates the task as the problem of detecting positive Fisher divergence between the noised null distribution and the noised, unknown data distribution. Diffusion models were initially proposed for generation since noising simplifies sampling, but they pose a conceptual puzzle in the context of hypothesis testing: the null and alternative hypotheses become harder to distinguish as the noise level increases. Therefore, aside from testing in Fisher divergence, diffusion models may face serious limitations in addressing fundamental hypothesis testing problems, such as testing in total variation distance. In this paper, we set out to rigorously characterize the statistical limits of diffusion's score-based approach to testing. We derive the minimax rate of testing in Fisher divergence against a broad alternative hypothesis consisting of densities which are compactly supported and assumed only to be bounded below by a constant. Notably, we capture the sharp scaling with respect to the the noise level. We then turn to testing in total variation, and since it is folklore that the problem is trivial without any regularity conditions, we study Hölder-smooth alternatives. As established in the literature, the Fisher divergence can be aggregated over noise levels to bound the total variation distance; hence, separation in total variation implies separation in aggregated Fisher divergence. After sharpening our Fisher divergence testing results to incorporate the available smoothness, we show that an aggregation of test statistics furnishes a test which achieves the sharp minimax testing rate in total variation. Hence, diffusion models are optimal for hypothesis testing.

## 1 Introduction

Recent advances in generative modeling have demonstrated the remarkable ability of diffusion models to capture complex probability distributions with high fidelity. These models operate on a simple idea. Clean training data are incrementally corrupted through a noising mechanism with a gradually increasing noise scale (often a diffusion process (Sohl-Dickstein et al., 2015; Ho et al., 2020)). At each step, a denoiser is learned to recover the current, noisy image from its noisier counterpart obtained at the next step. After the completion of all forward steps, the fitted denoiser has been learned to progressively denoise at each step; it is deployed on a fresh draw of pure noise to ultimately convert it into a new sample which is drawn approximately from the original, ground truth data-generating distribution. Central to this process is the estimation of the score function of the forward process. Through score matching (Hyvärinen, 2005; Vincent, 2011; Song & Ermon, 2019), diffusion models learn to approximate the true distribution of the data, enabling the generation of realistic samples from an unknown target distribution.

Besides estimation, hypothesis testing is an equally fundamental problem in statistical inference. Diffusion models are especially relevant for hypothesis testing from two angles. First, diffusion models are particularly flexible and successfully recover fine-grained details necessary for high-quality generation. This flexibility suggests the capacity to detect rare, subtle, and weak signals; big gains in power might be had by employing diffusion-based testing methodology. In fact, an emerging body of work has empirically shown better performance compared to testing methodology based on earlier generative frameworks (such as generative adversarial networks (GANs) and variational

autoencoders (VAEs)) (Wyatt et al., 2022; Bandara et al., 2022; Pinaya et al., 2022; Wolleb et al., 2022).

Second, with the widespread use of generative models, there are settings where the null hypothesis itself is some pre-trained diffusion model. For example, consider the extremely topical problem of detecting whether a given dataset of images has been generated from some specific AI model; this can be formulated as a hypothesis testing problem with that AI model specifying the null hypothesis. The problem is exceptionally important in the *post-hoc* regime where the model provider has not implemented any kind of reliable watermarking scheme. The same hypothesis testing formulation can address the problem of detecting distribution shift. A model provider may wish to update their specific diffusion model if the target distribution of natural, authentic images has shifted (Heng et al., 2024; Graham et al., 2023). Generative models have also been increasingly deployed in scientific applications (Angelopoulos et al., 2023; Wang et al., 2023). Generative models are especially attractive in domains where gold-standard experimental data is cost-prohibitive to collect.

Research efforts in diffusion models have largely been concentrated in practical, engineering aspects for the purposes of generation; essentially, the estimation problem has received most of the attention. The hypothesis testing context has been much less studied, and so even basic theoretical questions remain open. In this work, we address questions concerning statistical optimality in the context of applying diffusion models for hypothesis testing. Our focus is exclusively on the statistical aspect, and we leave computational and algorithmic considerations for future work.

## 1.1 BACKGROUND ON DIFFUSION MODELS

Before formulating the testing problem and the diffusion-based approach we are going to study, we first review some background on diffusion models. Consider a probability density function $f$ on $\mathbb{R}$ representing the unknown target distribution from which we wish to produce a sample[1]. Diffusion models approach this generation problem by considering a *forward process* and a *reverse process*, which are solutions to some specified stochastic differential equations (SDEs). Consider a generic SDE which implicitly defines the forward process $\{X_t\}_{t \geq 0}$ as its solution,

$$dX_t = g(X_t, t)\, dt + \sigma(t)\, dW_t, \quad X_0 \sim f \tag{1}$$

where $g : \mathbb{R} \times [0, \infty) \to \mathbb{R}$ is the drift, $\sigma : [0, \infty) \to [0, \infty)$ is the diffusion coefficient, and $\{W_t\}_{t \geq 0}$ is the standard Wiener process in $\mathbb{R}$. The practitioner chooses $g$ and $\sigma$ as part of their model design. For a fixed $T > 0$, the reverse process is defined as $Y_t := X_{T-t}$ for $0 \leq t \leq T$. Under some mild conditions, it is known (Anderson, 1982) that $\{Y_t\}_{0 \leq t \leq T}$ solves the following SDE,

$$dY_t = \left(-g(Y_t, T - t) + \sigma^2(T - t)s(Y_t, T - t)\right) dt + \sigma(T - t)\, dW_t, \quad Y_0 \sim p(\cdot, T) \tag{2}$$

where $p(\cdot, t)$ denotes the density of the marginal distribution of $X_t$, and the function $s(x, t) := \partial_x \log p(x, t)$ is referred to as the *score function* of the density $p(\cdot, t)$.

These probabilistic facts have algorithmic utility. For various choices of $g$ and $\sigma$ in (1), the distribution $p(\cdot, T)$ is close to some known distribution for large $T$ even though the initialization $f$ is not known. For example, if $g(x, t) = -x$ and $\sigma(t) = \sqrt{2}$, then $p(\cdot, T)$ is close to $N(0, 1)$ (this is referred to as the *variance preserving SDE* (Song et al., 2021)). Therefore, the practitioner can plug in an estimator of the score $\hat{s}(x, t)$ (constructed from available training data drawn from $f$) into (2) and solve the reverse SDE by using a known initialization rather than the unknown $p(\cdot, T)$ to obtain an approximate reverse process $\{\hat{Y}_t\}_{0 \leq t \leq T}$. Since the true reverse process satisfies $Y_T \sim f$, it is hoped that $\hat{Y}_T$ has a distribution which approximates $f$ well. The sources of error are the approximation of the initialization and of the score function. The former should be negligible since $p(\cdot, T)$ is close to a known distribution by design. The real challenge is to estimate the score function well (Oko et al., 2023; Dou et al., 2024; Zhang et al., 2024). For a detailed review of diffusion models and recent advances, we refer the reader to (Chen et al., 2024; Tang & Zhao, 2024; Yang et al., 2023).

---

[1]To maintain focus on the mathematical essence, we will work in the one-dimension setting so as not to get distracted by tedious, notational burdens in the multidimensional setting. In principle, the paper's results can be straightforwardly generalized without conceptual difficulty to the setting where the dimension is fixed and greater than one.

## 1.2 Hypothesis testing

The diffusion model framework focuses on the score function as the central statistical object. For the generation problem, estimation of the score function at each noise level $t$ is the chief task. For hypothesis testing, detecting deviations with respect to the Fisher divergence at each level $t$ becomes the task at hand. To elaborate, consider independent and identically distributed data $\mu_1, ..., \mu_n \sim f$ from an unknown distribution $f$. Further consider testing the null hypothesis $H_0 : f = f_0$, where $f_0$ is some known, reference distribution. The score-based paradigm of diffusion models casts the null hypothesis equivalently as $H_0 : \mathbb{F}_t(f \,||\, f_0) = 0$. Here, $t > 0$ and $\mathbb{F}_t$ denotes the Fisher divergence

$$\mathbb{F}_t(f \,||\, f_0) := \int_{-\infty}^{\infty} |s(x,t) - s_0(x,t)|^2 \, p_0(x,t) \, dx, \tag{3}$$

where $p_0(x,t)$ is the law of $X_t$ in the forward process $\{X_t\}_{t \geq 0}$ with initialization $f_0$ in (1) and $s_0(x,t) = \partial_x \log p_0(x,t)$ is its score function. The formulation in terms of the Fisher divergence immediately delineates the alternative hypothesis; the goal is to detect deviations with respect to the Fisher divergence. Concretely, for each $t > 0$ we have

$$\begin{aligned} H_0 &: \mathbb{F}_t(f \,||\, f_0) = 0, \\ H_1 &: \mathbb{F}_t(f \,||\, f_0) \geq \varepsilon_t^2 \text{ and } f \in \mathcal{F}. \end{aligned} \tag{4}$$

Here, $\mathcal{F}$ denotes a class of signals to be detected.

When speaking about the diffusion model approach to hypothesis testing, we are referring to the conceptual approach of viewing the testing problem (4) as the target problem to solve. While there may be many possible methodological strategies for addressing (4), understanding the fundamental hypothesis testing capability of diffusion models requires characterizing the statistical limits of the problem (4), which so far has been lacking in the literature. In this paper, we take a minimax perspective on statistical limits. Namely, we denote $\varepsilon_t^*$ as the minimax testing rate (also referred to as the *minimax separation rate*) for (4); a rigorous definition is given in Appendix A. The following fundamental question immediately arises.

**Question 1.** *For any $t > 0$, what is the minimax testing rate for (4)?*

An answer to Question 1 sharply describes the testing capability of diffusion models by precisely characterizing the magnitude of deviations in the alternative hypothesis (4) which are necessary and sufficient for successful detection. To probe diffusion model's capacity for hypothesis testing in the most general setting, we will take $\mathcal{F}$ to be very broad in the first part of the paper, namely densities which are only assumed to be supported on $[-1, 1]$ and bounded below by a constant. Later on, we will also investigate whether the diffusion approach has the capability to exploit smoothness, which is particularly important for small $t$; we will consider a class of Hölder-smooth signals.

The diffusion approach to hypothesis testing poses an interesting statistical puzzle. From the estimation perspective, the score function $s(x,t)$ becomes easier to estimate as $t$ increases since the target density $p(x,t)$ looks more and more like noise; estimation of pure noise is a trivial problem. This progressive simplification of the estimation problem is the basic premise of diffusion from the generation vantage point. However, it is not immediately clear this is helpful in the testing problem. As $t$ gets larger and larger, *both* the noised null $p_0(x,t)$ and the noised data density $p(x,t)$ look like noise, which is to say both distributions become more and more indistinguishable. In other words, noising seems to make the testing problem harder! It appears a progressively larger and larger signal is needed to distinguish the two.

This statistical puzzle might also suggest that the diffusion model approach may face serious limitations in addressing foundational hypothesis testing problems. Concretely, consider testing in total variation distance,

$$\begin{aligned} H_0 &: f = f_0, \\ H_1 &: \mathrm{d}_{\mathrm{TV}}(f, f_0) \geq \varepsilon \text{ and } f \in \mathcal{F}. \end{aligned} \tag{5}$$

The problem (5) is clearly of fundamental interest and is formulated from first principles without any reference to a particular approach. The minimax rate $\varepsilon^*$ for (5) can be defined analogously to (1). The problem (5) is related to (4) by the following well-known bound (Oko et al., 2023; Chen et al., 2023),

$$\mathrm{d}_{\mathrm{TV}}(f, f_0)^2 \lesssim \mathrm{d}_{\mathrm{KL}}(f * \varphi_T \,||\, f_0 * \varphi_T) + \int_0^T \mathbb{F}_t(f \,||\, f_0) \, dt, \tag{6}$$

For large $T$ the first term is negligible, so let us ignore it for now. The bound (6) implies that a separation $d_{\text{TV}}(f, f_0) \geq \varepsilon$ in the alternative hypothesis implies that the signals in (4) are large after *aggregating over* $t$, namely we have $\varepsilon^2 \lesssim \int_0^T \mathbb{F}_t(f \| f_0) \, dt$. An essential question is whether the diffusion model approach can optimally solve (5) by aggregating the problems (4) over $t$.

**Question 2.** *If (4) can be optimally tested at every noise level $t$, can the problem (5) be tested at the minimax rate by aggregating the evidence across $t$?*

The testing problem (5) has been extensively studied in the nonparametric statistics literature (e.g. see (Ingster & Suslina, 2003; Giné & Nickl, 2016) and references therein). For our first choice of $\mathcal{F}$ which imposes essentially no assumptions, it is well known the problem (5) is trivial (i.e. testing is impossible even with constant order $\varepsilon$). Constraints are necessary for (5) to be meaningful. For various classical choices of $\mathcal{F}$ (e.g. which impose smoothness assumptions), minimax rates have long been established in the literature and are well understood. For our later choice of Hölder-smooth $\mathcal{F}$, the point of Question 2 is to determine whether the diffusion approach (i.e. the approach which targets (4)) can yield an optimal test for (5) and match the known minimax rate.

## 1.3 Main contributions

To focus on core ideas, we focus on the diffusion model associated to the *variance exploding SDE* (Song et al., 2021) in (1) given by $g \equiv 0$ and $\sigma \equiv 1$. Namely, the forward process $\{X_t\}_{t \geq 0}$ solves

$$dX_t = dW_t, \quad X_0 \sim f. \tag{7}$$

Note the marginal distribution of $X_t$ has density $p(x, t) = (f * \varphi_t)(x)$ where $*$ denotes convolution and $\varphi_t(x) = \frac{1}{\sqrt{2\pi t}} e^{-\frac{x^2}{2t}}$ is the density of $N(0, t)$. The corresponding reverse process $\{Y_t\}_{0 \leq t \leq T}$ solves (2), which is given by

$$dY_t = s(Y_t, T - t) \, dt + dW_t, \quad Y_0 \sim p(\cdot, T). \tag{8}$$

Recall $Y_T \sim f$ and $s(x, t) = \partial_x \log p(x, t) = \frac{\partial_x p(x, t)}{p(x, t)}$.

In the hypothesis testing problem (4), we will look at a very broad class of alternative densities with the goal of understanding diffusion's general capacity for testing. Define

$$\mathcal{F} := \left\{ f : \mathbb{R} \to \mathbb{R} : \text{supp}(f) \subset [-1, 1], \int_{-\infty}^{\infty} f(\mu) \, d\mu = 1, \text{ and } c_d \leq f(\mu) \leq C_d \text{ for all } |\mu| \leq 1 \right\} \tag{9}$$

where $C_d, c_d > 0$ are some universal constants. Moving to (5), it is well known that the choice of $\mathcal{F}$ yields triviality. Moreover, it is interesting to ask whether the diffusion approach to testing can exploit available smoothness in the problem. With this motivation, we will also consider the setting of Hölder smoothness. For any (possibly unbounded) interval $A \subset \mathbb{R}$, define the Hölder space

$$\mathcal{H}_\alpha(A; L) := \left\{ f : \mathbb{R} \to \mathbb{R} : \int_{-\infty}^{\infty} f(\mu) \, d\mu = 1, \text{supp}(f) \subset A, \max_{0 \leq j \leq \lfloor \alpha \rfloor} \sup_{\mu \in A} |f^{(j)}(\mu)| \leq L, \text{ and} \right.$$

$$\left. |f^{(\lfloor \alpha \rfloor)}(\mu) - f^{(\lfloor \alpha \rfloor)}(\mu')| \leq L|\mu - \mu'|^{\alpha - \lfloor \alpha \rfloor} \text{ for all } \mu, \mu' \in A \right\}.$$

We will focus on the setting where $L$ is a small universal constant, and we will suppress it from notation. We also treat $\alpha > 0$ as a fixed constant and all explicit or implicit universal constants may depend on it. The class of signals $f$ in the alternative will be taken to be densities supported on $[-1, 1]$, $\alpha$-Hölder on their supports, and bounded below by a constant. Furthermore, we will assume the difference $f - f_0$ is $\alpha$-Hölder on $\mathbb{R}$. Define the parameter space

$$\mathcal{F}_\alpha(L) := \{ f \in \mathcal{F} : f \in \mathcal{H}_\alpha([-1, 1]; L) \text{ and } f - f_0 \in \mathcal{H}_\alpha(\mathbb{R}; L) \}. \tag{10}$$

Throughout the paper, both in studying $\mathcal{F}$ and $\mathcal{F}_\alpha$, we will make the following assumption about $f_0$.

**Assumption 1.** *Assume $f_0 \in \mathcal{H}_{\alpha \vee 1}([-1, 1]; L)$, $f_0(-1) = f_0(1)$, and $c_d \leq f_0(\mu) \leq C_d$ for all $\mu \in [-1, 1]$ where $C_d, c_d > 0$ are some universal constants.*

Besides being bounded below by a constant (a standard assumption (Tsybakov, 2009)), Assumption 1 essentially assumes $f_0$ has at least one bounded derivative and is periodic. From an information-theoretic point of view, this assumption is actually without loss of generality, since it is assumed

$f_0$ is known. By simply transforming the data by the inverse c.d.f. transform, the transformed data follow the uniform distribution on $[-1, 1]$ (after an additional transformation to map $[0, 1]$ to $[-1, 1]$). In other words, one can take $f_0 = \frac{1}{2}\mathbb{1}_{\{|\mu| \leq 1\}}$ without loss of generality, which clearly lives in $\mathcal{H}_{\alpha \vee 1}(L)$ and is periodic. This reduction to the uniform case is standard (Giné & Nickl, 2016; Ingster, 1994; 2000). The periodicity assumption will be used as we will analyze estimators based on Fourier series projection ideas.

Our first main contribution is an answer to Question 1, namely that the minimax testing rate for (4) with the choice $\mathcal{F}$ given by (9) is

$$\varepsilon_t^*(\mathcal{F})^2 \asymp \frac{1}{nt^2} \wedge \frac{1}{nt^{5/4}} \wedge \frac{1}{t}. \tag{11}$$

The minimax rate (11) turns out to decrease in $t$, meaning that detection actually becomes easier as the noise level increases, which apparently contradicts the intuition that $f * \varphi_t$ and $f_0 * \varphi_t$ ought to get progressively more indistinguishable. Though appealing, this intuition neglects the fact that the noised densities become smoother as $t$ increases. Consequently, larger $t$ has some statistical benefit since more regularity is available to exploit in the target $\mathbb{F}_t(f \,\|\, f_0)$. It turns out these two competing effects balance out in such a way to yield (11).

A natural idea for testing (4) is to furnish an estimator $\widehat{\mathbb{F}}_t$ of the Fisher divergence $\mathbb{F}_t$ and use it as a test statistic, rejecting the null hypothesis when it exceeds some threshold. This strategy is employed in the very high noise regime $t \gtrsim 1$. However, in the regime $t \lesssim 1$, it turns out constructing an estimator of $\mathbb{F}_t$ which is optimal (especially having error with sharp dependence on $t$) is challenging. Instead, suppose we had an upper bound $\mathbb{F}_t(f \,\|\, f_0) \leq \mathbb{U}_t(f \,\|\, f_0)$ which satisfies $\mathbb{U}_t(f \,\|\, f_0) = 0$ under the null hypothesis $f = f_0$. Then, under the alternative hypothesis, we have $\varepsilon_t^2 \leq \mathbb{U}_t(f \,\|\, f_0)$, which is to say the alternative hypothesis is also separated away from the null in terms of $\mathbb{U}_t$, not just $\mathbb{F}_t$. Then it is an appealing idea to construct an estimator $\widehat{\mathbb{U}}_t$ for the proxy $\mathbb{U}_t$ and use it as a test statistic, especially if it is easier to estimate than $\mathbb{F}_t$. Of course, the difficulty should not be greater than the target rate $\varepsilon_t^*(\mathcal{F})^2$ given by (11), and typically this means $\mathbb{U}_t$ should be a fairly close upper bound. Our methodology implements this strategy, and different choices of the upper bound are made in different regimes of $t$.

Our second contribution is an affirmative answer to Question 2 with the choice of $\mathcal{F}_\alpha$ given by (10) in the alternative hypothesis. The bound (6) suggests an appealing aggregation idea for furnishing a test statistic. The Kullback-Leibler divergence term is negligible as it is of order at most $\frac{1}{T}$ and $T$ will be chosen large (say, $T \asymp n$), so let us ignore it. Therefore, under the alternative hypothesis $d_{\mathrm{TV}}(f, f_0)^2 \geq \varepsilon^2$, it follows $\varepsilon^2 \lesssim \int_0^T \mathbb{F}_t(f \,\|\, f_0)\, dt \leq \int_0^T \mathbb{U}_t(f \,\|\, f_0)\, dt$. Intuitively, the right-hand side can be estimated at rate $\int_0^T \frac{1}{nt^2} \wedge \frac{1}{nt^{5/4}} \wedge \frac{1}{t}\, dt$ by the aggregated statistic $\int_0^T \widehat{\mathbb{U}}_t\, dt$.

However, it quickly becomes clear that the estimators $\widehat{\mathbb{U}}_t$ developed for $\mathcal{F}$ instead of $\mathcal{F}_\alpha$ will not be optimal. Specifically, the bound $\int_0^T \frac{1}{nt^2} \wedge \frac{1}{nt^{5/4}} \wedge \frac{1}{t}\, dt$ is problematic since $\int_0^{n^{-4}} \frac{1}{t}\, dt = \infty$. The error bound for small $t$ is not good enough, and one might hope to do better by exploiting the smoothness in $\mathcal{F}_\alpha$ rather than using estimators developed for $\mathcal{F}$. Such blow-up issues with (6) frequently occur in the diffusion literature, and the very popular early stopping is used to circumvent this problem. However, early stopping introduces logarithmic factors which may be suboptimal; for example, Dou et al. (2024) found early stopping is not at all necessary to achieve the optimal density estimation rate.

The testing rate can be improved by leveraging the smoothness, and we show that when $t \leq c$ for a sufficiently small constant $c$, the minimax testing rate of (4) with $\mathcal{F}_\alpha$ is given by

$$\varepsilon_t^*(\mathcal{F}_\alpha)^2 \asymp \frac{1}{nt^{5/4}} \wedge \left( n^{-\frac{4(\alpha-1)}{4\alpha+1}} + t^{\alpha-1} \right). \tag{12}$$

Importantly, when $\alpha \geq 1$, we have $n^{-\frac{4(\alpha-1)}{4\alpha+1}} + t^{\alpha-1} \asymp n^{-\frac{4(\alpha-1)}{4\alpha+1}}$, which does not blow up when integrating! Likewise, when $\alpha < 1$, this regime specializes to $t^{\alpha-1}$, which is also integrable over small $t$. Consequently, we are able to show that the aggregation statistic is able to achieve the known, minimax testing rate for (5), which is $(\varepsilon^*)^2 \asymp n^{-\frac{4\alpha}{4\alpha+1}}$ and was proved by Ingster & Suslina (2003). Hence, we can conclude diffusion models are optimal for hypothesis testing in the context of Hölder-smooth signals.

## 2 TESTING IN FISHER DIVERGENCE

In this section, we address the upper bound aspect Question 1, namely the question of testing (4) with the parameter space $\mathcal{F}$ given by (9). We defer the minimax lower bound to Appendix C.

As mentioned in Section 1.3, in the very high noise regime $t \gtrsim 1$ an optimal test can be constructed by plugging in an optimal score estimator to estimate $\mathbb{F}_t$. As described earlier, a different idea is employed in the $t \lesssim 1$ regime. We furnish a population quantity which upper bounds $\mathbb{F}_t$ and is also equal to zero under the null hypothesis. Estimators for this proxy quantity are constructed and used as test statistics for testing (4).

### 2.1 VERY HIGH NOISE REGIME

In the very high noise regime $t \gtrsim 1$, an optimal test is constructed via estimating the Fisher divergence $\mathbb{F}_t(f \,\|\, f_0)$ by an estimator which plugs in a score estimator. The optimal score estimator from Dou et al. (2024) is used, which we reproduce here. The estimator is motivated by the following representation of the score $s(x,t) = \frac{\partial_x p(x,t)}{p(x,t)}$. Note that the partial derivatives are given by $\partial_x p(x,t) = \partial_x \int_{-1}^{1} \varphi_t(x-\mu) f(\mu) \, d\mu = \int_{-1}^{1} -\frac{x-\mu}{t} \varphi_t(x-\mu) f(\mu) \, d\mu = -\frac{x}{t} p(x,t) + \frac{1}{t} \int_{-1}^{1} \mu \varphi_t(x-\mu) f(\mu) \, d\mu$. Therefore, we have $\partial_x p(x,t) = -\frac{x}{t} p(x,t) + \frac{1}{t} \int_{-1}^{1} \mu \varphi_t(x-\mu) f(\mu) \, d\mu$. This representation gives us a way to estimate $\partial_x p(x,t)$ from $p(x,t)$. Let $\varepsilon(x,t) := c_d \int_{-1}^{1} \varphi_t(x-\mu) \, d\mu$ and note $p(x,t) \geq \varepsilon(x,t)$ for all $f \in \mathcal{F}_\alpha$. Define $\hat{p}(x,t) := \varepsilon(x,t) \vee \frac{1}{n} \sum_{i=1}^{n} \varphi_t(x-\mu_i)$ and $\widehat{\partial_x p}(x,t) := -\frac{x}{t} \hat{p}(x,t) + \frac{1}{nt} \sum_{i=1}^{n} \mu_i \varphi_t(x-\mu_i)$. The score estimator of Dou et al. (2024) is $\hat{s}(x,t) := \frac{\widehat{\partial_x p}(x,t)}{\hat{p}(x,t)}$. The score estimator $\hat{s}$ is plugged in to obtain an estimator of the Fisher divergence, $\widehat{\mathbb{F}}_t := \int_{-\infty}^{\infty} |\hat{s}(x,t) - s_0(x,t)|^2 \, p_0(x,t) \, dx$.

We use the test

$$\phi_t := \mathbb{1}\left\{ \widehat{\mathbb{F}}_t \geq \frac{C_\eta'}{nt^2} \right\} \tag{13}$$

where $C_\eta'$ is a constant to be tuned to achieve a testing risk of at most $\eta$.

**Theorem 1.** *If $t \geq 1$ and $\eta > 0$, then there exist $C_\eta, C_\eta' > 0$ depending only on $\eta$ such that for all $C > C_\eta$, we have*

$$P_{f_0}\{\phi_t = 1\} + \sup_{\substack{f \in \mathcal{F}, \\ \mathbb{F}_t(f \,\|\, f_0) \geq C \varepsilon_t^2}} P_f\{\phi_t = 0\} \leq \eta,$$

*where $\varepsilon_t^2 = \frac{1}{nt^2}$ and $\phi_t$ is given by (13).*

The proof of Theorem 1 is deferred to Appendix B.1. The argument is straightforward, and relies on the error bound of the plugged-in score estimator $\hat{s}$ provided by Dou et al. (2024).

### 2.2 HIGH NOISE REGIME

Outside the regime $t \gtrsim 1$, the proxy strategy described in Section 1.3 and at the beginning of Section 2 is employed. Proposition 1 states the proxy we will target in the high noise regime $n^{-4} \lesssim t < 1$. Its proof is deferred to Appendix B.2.

**Proposition 1.** *We have $\mathbb{F}_t(f \,\|\, f_0) \lesssim Q_t + Q_t'$ where*

$$Q_t = \int_{-\infty}^{\infty} \frac{|\partial_x p_0(x,t)|^2}{p_0(x,t)^3} |p(x,t) - p_0(x,t)|^2 \, dx, \tag{14}$$

$$Q_t' = \int_{-\infty}^{\infty} \frac{|\partial_x p(x,t) - \partial_x p_0(x,t)|^2}{p_0(x,t)} \, dx. \tag{15}$$

Define the estimators

$$\hat{Q}_t = \frac{1}{\binom{n}{2}} \sum_{i \neq j} \int_{-\infty}^{\infty} \frac{|\partial_x p_0(x,t)|^2}{p_0(x,t)^3} (\varphi_t(x - \mu_i) - p_0(x,t))(\varphi_t(x - \mu_j) - p_0(x,t)) \, dx, \quad (16)$$

$$\hat{Q}_t' = \frac{1}{\binom{n}{2}} \sum_{i \neq j} \int_{-\infty}^{\infty} \frac{(\partial_x \varphi_t(x - \mu_i) - \partial_x p_0(x,t))(\partial_x \varphi_t(x - \mu_j) - \partial_x p_0(x,t))}{p_0(x,t)} \, dx. \quad (17)$$

The test

$$\phi_t = \mathbb{1}\left\{ |\hat{Q}_t + \hat{Q}_t'| \geq \frac{C_\eta'}{nt^{5/4}} \right\} \quad (18)$$

is employed, where $C_\eta'$ is a constant to be tuned to achieve a testing risk of at most $\eta$.

**Theorem 2.** *Suppose $t < 1$. If $\eta > 0$, then there exist $C_\eta, C_\eta' > 0$ depending only on $\eta$ such that for all $C > C_\eta$, we have*

$$P_{f_0}\{\phi_t = 1\} + \sup_{\substack{f \in \mathcal{F}, \\ \mathbb{F}_t(f \,||\, f_0) \geq C\varepsilon_t^2}} P_f\{\phi_t = 0\} \leq \eta,$$

*where $\varepsilon_t^2 = \frac{1}{nt^{5/4}}$ and $\phi_t$ is given by (18).*

The proof is deferred to Appendix B.2.1. The result of Theorem 2 holds for all $t < 1$, but is only relevant in the high noise regime $n^{-4} \lesssim t < 1$ where it achieves the minimax rate (11). The low noise regime requires a different approach. The argument of Theorem 2 relies on sharp calculations of the variances of the $U$-statistics $\hat{Q}_t$ and $\hat{Q}_t'$.

## 2.3 LOW NOISE REGIME

In the low noise regime $t \lesssim n^{-4}$, the rate (11) specializes to $t^{-1}$ for $\alpha < 1$. The trivial test which always accepts the null hypothesis achieves the separation rate $t^{-1}$. Proposition 4 shows the reason for the triviality, and is an immediate corollary of Lemma 3.1 of (Gupta et al., 2022), which states a bound on the Fisher information of $f * \varphi_t$, namely $\int_{-\infty}^{\infty} |s(x,t)|^2 \, p(x,t) \, dx \lesssim t^{-1}$.

**Proposition 2.** *We have $\sup_{f \in \mathcal{F}} \mathbb{F}_t(f \,||\, f_0) \lesssim t^{-1}$.*

*Proof.* Fix $f \in \mathcal{F}$. Since $f_0 \asymp f$, we have from the inequality $(a + b)^2 \lesssim a^2 + b^2$ that $\mathbb{F}_t(f \,||\, f_0) \lesssim \int_{-\infty}^{\infty} |s(x,t)|^2 \, p(x,t) \, dx + \int_{-\infty}^{\infty} |s_0(x,t)|^2 \, p_0(x,t) \, dx$. Lemma 3.1 of (Gupta et al., 2022) immediately delivers the claim. $\square$

Proposition 4 shows that the diameter of the entire alternative hypothesis is of order at most $t^{-1}$. Consequently, for a sufficiently large universal constant $C^* > 0$, there does not exist any $f \in \mathcal{F}$ such that $\mathbb{F}_t(f \,||\, f_0) \geq C^* t^{-1}$. Consequently, the test which always accepts the null hypothesis achieves exactly zero testing risk for testing $H_0 : \mathbb{F}_t(f \,||\, f_0) = 0$ against $H_1 : \mathbb{F}_t(f \,||\, f_0) \geq C t^{-1}$ and $f \in \mathcal{F}$ for all $C > C^*$. Hence, it achieves the separation rate $t^{-1}$. This result is formally stated in Theorem 3 without proof.

**Theorem 3.** *There exists a universal constant $C^* > 0$ such that for all $C > C^*$, we have $P_{f_0}\{\phi_0 = 1\} + \sup_{\substack{f \in \mathcal{F}, \\ \mathbb{F}_t(f \,||\, f_0) \geq C\varepsilon_t^2}} P_f\{\phi_0 = 0\} = 0$, where $\varepsilon_t^2 = t^{-1}$ and $\phi_0 \equiv 0$.*

Theorem 3 holds for all $t > 0$, but is only relevant in the regime $t \lesssim n^{-4}$ to deliver the upper bound in (11).

## 3 TESTING IN TOTAL VARIATION DISTANCE

In this section, we address testing (5) with the parameter space $\mathcal{F}_\alpha$ given by (10).

### 3.1 EXPLOITING SMOOTHNESS TO IMPROVE TESTING IN FISHER DIVERGENCE

As mentioned in Section 1.3, the key step is to improve the Fisher divergence testing rate (11) to (12) for small $t$. This section thus describes the upper bound aspect of the optimal minimax testing rate $\varepsilon_t^*(\mathcal{F}_\alpha)$ given by (12). The lower bound is deferred to Appendix C.

From the upper bound perspective and in view of (12), we only need to improve in the regime $t \lesssim n^{-4/(4\alpha+1)}$ with the target $n^{-4(\alpha-1)/(4\alpha+1)} + t^{\alpha-1}$. Note this target specializes to $n^{-\frac{4(\alpha-1)}{4\alpha+1}}$ for $\alpha \geq 1$ and to $t^{\alpha-1}$ for $\alpha < 1$. These two cases are handled separately. The tests of Sections 2.1 and 2.2 can be immediately used to address the regime $t \gtrsim 1$ and $n^{-4(\alpha-1)/(4\alpha+1)} \lesssim t < 1$ respectively.

Suppose $\alpha \geq 1$. The proxy strategy is again employed, though the target estimand is different. The test (18) is not appropriate in the low noise regime, since it achieves a separation rate $\frac{1}{nt^{5/4}}$ which diverges as $t \to 0$. To achieve the desired rate $n^{-4(\alpha-1)/(4\alpha+1)} + t^{\alpha-1}$, which notably does not diverge, the smoothness of $f$ and $f_0$ will be explicitly exploited. The proxy given in Proposition 3 is useful precisely because of the availability of smoothness.

**Proposition 3.** *If $\alpha \geq 1$ and $t < 1$, then $\mathbb{F}_t(f \parallel f_0) \lesssim Q + Q' + t^{\alpha-1}$ where $Q = \int_{-1}^1 |f(\mu) - f_0(\mu)|^2 \, d\mu$ and $Q' = \int_{-1}^1 |f'(\mu) - f_0'(\mu)|^2 \, d\mu$.*

The proof of Proposition 3 is deferred to Appendix E.1. The proxy $Q + Q'$ will be our target estimand. The intuition is that for small $t$, the Fisher divergence $\mathbb{F}_t(f \parallel f_0)$ ought to be close to $\mathbb{F}_0(f \parallel f_0)$, that is when $t = 0$, due to the smoothness of the underlying densities. At $t = 0$, since $f_0 \asymp f \asymp 1$, it is very natural to obtain a proxy estimand as follows,

$$\mathbb{F}_0(f \parallel f_0) = \int_{-1}^1 \frac{|f'(\mu)f_0(\mu) - f_0'(\mu)f(\mu)|^2}{f_0(\mu)^2 f(\mu)^2} f_0(\mu) \, d\mu \lesssim Q + Q'.$$

When $t > 0$, some error will be incurred and will imply some conditions on how small $t$ must be. To achieve the optimal rate $n^{-\frac{4(\alpha-1)}{4\alpha+1}}$, it turns out the error $t^{\alpha-1}$ in Proposition 3 is only negligible when $t \lesssim n^{-\frac{4}{4\alpha+1}}$, which is exactly the low noise regime we are considering!

The estimation of $Q$ and $Q'$ is classical (Bickel & Ritov, 1988; Laurent, 1996; Giné & Nickl, 2008). We employ the estimators based on orthogonal series (Laurent, 1996). Formally, let $\{\psi_k\}_{k=1}^\infty$ denote the usual trigonometric basis in $L^2([-1,1])$ with $\psi_1(y) = \frac{1}{\sqrt{2}}$, $\psi_{2k}(y) = \cos(\pi k(y+1))$ and $\psi_{2k+1}(y) = \sin(\pi k(y+1))$. Since $f$ and $f_0$ are periodic on $[-1,1]$, we have the basis expansions $f_0 = \sum_{k=1}^\infty \theta_{0,k}\psi_k$ and $f = \sum_{k=1}^\infty \theta_k\psi_k$. Moreover, we have estimates on the coefficient decay. Define the ellipsoid $\Theta_\alpha(L) = \{\theta \in \ell^2(\mathbb{N}) : \sum_{k=1}^\infty a_k^2\theta_k^2 \leq L\}$ where $a_k = k^\alpha$ if $k$ is even and $a_k = (k-1)^\alpha$ if $k$ is odd. We have by standard results (Tsybakov, 2009) that the basis coefficients of $f$ and $f_0$ live in $\Theta_\alpha(L)$. By Parseval's identity, we have $Q = \sum_{k=1}^\infty (\theta_k - \theta_{0,k})^2$, and so our estimator is defined to be

$$\hat{Q}_K := \frac{1}{\binom{n}{2}} \sum_{i \neq j} \sum_{k=1}^K (\psi_k(\mu_i) - \theta_{0,k})(\psi_k(\mu_j) - \theta_{0,k}). \tag{19}$$

Here, $K$ is a tuning parameter.

To estimate $f' = \sum_{k=1}^K \theta_k\psi_k'$, consider $\psi_{2k}' = -\pi k\psi_{2k+1}$ and $\psi_{2k+1}' = \pi k\psi_{2k}$ since $\{\psi_k\}_{k=1}^\infty$ is the trigonometric basis. Therefore, $f'(\mu) = \sum_{k=1}^\infty \pi k\theta_{2k+1}\psi_{2k}(\mu) - \pi k\theta_{2k}\psi_{2k+1}(\mu)$. Let us denote the basis coefficients of $f'$ by $\tilde{\theta}_{2k} := \pi k\theta_{2k+1}$ and $\tilde{\theta}_{2k+1} = -\pi k\theta_{2k}$. Likewise, let $\tilde{\theta}_{0,k}$ denote the corresponding coefficients of $f_0'$. Define the estimator

$$\hat{Q}_K' = \frac{1}{\binom{n}{2}} \sum_{i \neq j} \sum_{k=1}^K A_k(\mu_i)A_k(\mu_j), \tag{20}$$

where $A_k(\mu) := (\pi k\psi_{2k+1}(\mu) - \tilde{\theta}_{0,2k}) + (-\pi k\psi_{2k}(\mu) - \tilde{\theta}_{0,2k+1})$. The test

$$\phi := \mathbb{1}\left\{|\hat{Q}_K + \hat{Q}_K'| \geq C_\eta' n^{-\frac{4(\alpha-1)}{4\alpha+1}}\right\} \tag{21}$$

is used in this regime, where $C'_\eta > 0$ is a constant to be tuned to achieve a testing risk of at most $\eta$. Theorem 4 establishes $\phi$ is rate-optimal, and its proof is deferred to Appendix B.3.

**Theorem 4.** *Suppose $\alpha \geq 1$ and $t \leq n^{-\frac{4}{4\alpha+1}}$. If $\eta > 0$, then there exist $C_\eta, C'_\eta > 0$ depending only on $\eta$ such that for all $C > C_\eta$, we have*

$$P_{f_0}\{\phi = 1\} + \sup_{\substack{f \in \mathcal{F}_\alpha, \\ \mathbb{F}_t(f \,\|\, f_0) \geq C\varepsilon_t^2}} P_f\{\phi = 0\} \leq \eta,$$

*where $\varepsilon_t^2 = n^{-\frac{4(\alpha-1)}{4\alpha+1}}$ and $\phi$ is given by (21) with $K = \left\lceil n^{\frac{2}{4\alpha+1}} \right\rceil$.*

On the other hand, suppose $\alpha < 1$. It turns out the trivial test which always accepts the null hypothesis achieves the separation rate $t^{\alpha-1}$. Proposition 4 shows the reason for the triviality, and is an immediate corollary of Theorem 5 of (Dou et al., 2024).

**Proposition 4.** *If $\alpha < 1$ and $t \lesssim n^{-\frac{4}{4\alpha+1}}$, then $\sup_{f \in \mathcal{F}_\alpha} \mathbb{F}_t(f \,\|\, f_0) \lesssim t^{\alpha-1}$.*

*Proof.* Fix $f \in \mathcal{F}_\alpha$. Let $u(\mu) = \frac{1}{2}\mathbb{1}_{\{|\mu| \leq 1\}}$ denote the density of the uniform distribution on $[-1, 1]$. Let $p_u(x, t) = (u * \varphi_t)(x)$ and $s_u(x, t) = \partial_x \log p_u(x, t)$. Consider $n^{-\frac{4}{4\alpha+1}} \leq n^{-\frac{2}{2\alpha+1}}$, and so we can apply Theorem 5 of (Dou et al., 2024) to conclude $\mathbb{F}_t(u \,\|\, f_0) \lesssim t^{\alpha-1}$ and $\mathbb{F}_t(u \,\|\, f) \lesssim t^{\alpha-1}$. Since $p_0 \asymp p$ and $(a+b)^2 \lesssim a^2 + b^2$, it is immediate that $\mathbb{F}_t(f \,\|\, f_0) \lesssim \mathbb{F}_t(u \,\|\, f_0) + \mathbb{F}_t(u \,\|\, f) \lesssim t^{\alpha-1}$, as desired. $\square$

Proposition 4 shows that the diameter of the entire alternative hypothesis is of order at most $t^{\alpha-1}$, and so the trivial test achieves the separation rate $t^{\alpha-1}$. This result is formally stated in Theorem 5 without proof.

**Theorem 5.** *Suppose $\alpha < 1$ and $t \leq n^{-\frac{4}{4\alpha+1}}$. There exists a universal constant $C^* > 0$ such that for all $C > C^*$, we have $P_{f_0}\{\phi_0 = 1\} + \sup_{\substack{f \in \mathcal{F}_\alpha, \\ \mathbb{F}_t(f \,\|\, f_0) \geq C\varepsilon_t^2}} P_f\{\phi_0 = 0\} = 0$, where $\varepsilon_t^2 = t^{\alpha-1}$ and $\phi_0 \equiv 0$.*

## 3.2 AGGREGATION

The estimators we have constructed can be aggregated to furnish an optimal test for the classical problem (5) of detecting alternatives separated from the null hypothesis in total variation distance. Define

$$\widehat{\mathbb{U}}_t := \begin{cases} \widehat{\mathbb{F}}_t & \text{if } t \geq 1, \\ \hat{Q}_t + \hat{Q}'_t & \text{if } n^{-\frac{4}{4\alpha+1}} < t < 1, \\ \hat{Q}_K + \hat{Q}'_K & \text{if } t \leq n^{-\frac{4}{4\alpha+1}}, \end{cases} \tag{22}$$

where $\widehat{\mathbb{F}}_t$ is given in Section 2.1, $\hat{Q}_t$ and $\hat{Q}'_t$ are given by (16) and (17) respectively, and $\hat{Q}_K$ and $\hat{Q}'_K$ are given by (19) and (20) with $K = \lceil n^{\frac{2}{4\alpha+1}} \rceil$. For $T > 0$, define the test

$$\phi_T := \mathbb{1}\left\{ \int_0^T \widehat{\mathbb{U}}_t \, dt \geq C'_\eta n^{-\frac{4\alpha}{4\alpha+1}} \right\} \tag{23}$$

where $C'_\eta$ is a constant to be tuned to achieve a testing risk of at most $\eta$.

**Theorem 6.** *Suppose $T \gtrsim n$. If $\eta > 0$, then there exist $C_\eta, C'_\eta > 0$ depending only on $\eta$ such that for all $C > C_\eta$, we have*

$$P_{f_0}\{\phi_T = 1\} + \sup_{\substack{f \in \mathcal{F}_\alpha, \\ \mathrm{d}_{\mathrm{TV}}(f, f_0) \geq C\varepsilon^*}} P_f\{\phi_T = 0\} \leq \eta,$$

*where $\varepsilon^* = n^{-\frac{2\alpha}{4\alpha+1}}$ and $\phi_T$ is given by (23).*

Theorem 6 establishes that the minimax rate $(\varepsilon^*)^2 \asymp n^{-\frac{4\alpha}{4\alpha+1}}$ can be achieved. Intuitively, this is precisely because the estimation error at every $t$ integrates to exactly the desired rate; we have $\int_0^T \frac{1}{nt^2} \wedge \frac{1}{nt^{5/4}} \wedge \left( n^{-\frac{4(\alpha-1)}{4\alpha+1}} + t^{\alpha-1} \right) dt \lesssim n^{-\frac{4\alpha}{4\alpha+1}}$. In this sense, we conclude diffusion models are optimal for hypothesis testing.

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

# Appendices to "Diffusion models are optimal for hypothesis testing"

## A    PRELIMINARIES

Definition 1 defines the minimax testing rate for (4).

**Definition 1.** *For $t > 0$, we say $\varepsilon_t^*$ is the minimax testing rate for (4) if for all $\eta \in (0, 1)$,*

*(i) there exists $C_\eta > 0$ depending only on $\eta$ such that for all $C > C_\eta$, we have*

$$\inf_\phi \left\{ P_{f_0} \{\phi = 1\} + \sup_{\substack{f \in \mathcal{F}, \\ \mathbb{F}_t(f \,||\, f_0) \geq C(\varepsilon_t^*)^2}} P_f \{\phi = 0\} \right\} \leq \eta,$$

*(ii) there exists $c_\eta > 0$ depending only on $\eta$ such that for all $0 < c < c_\eta$, we have*

$$\inf_\phi \left\{ P_{f_0} \{\phi = 1\} + \sup_{\substack{f \in \mathcal{F}, \\ \mathbb{F}_t(f \,||\, f_0) \geq c(\varepsilon_t^*)^2}} P_f \{\phi = 0\} \right\} \geq 1 - \eta,$$

*where the infimums run over all tests $\phi$ (i.e. binary measurable functions taking the data as input).*

Item *(i)* in Definition 1 is the upper bound criterion, and item *(ii)* is the lower bound criterion. Note $\varepsilon_t^*$ characterizes the hardness of the testing problem only up to constant factors.

## B    PROOFS OF THE UPPER BOUNDS

### B.1    VERY HIGH NOISE: PROOF OF THEOREM 1

The proof of Theorem 1 relies on estimation error bounds of the plugged-in estimators. The test (13) plugs in the score estimator of Dou et al. (2024). This score estimator is rate-optimal for score estimation and achieves the following error bound. Note the parameter space we consider is a subset of the parameter space in (Dou et al., 2024), and so their upper bound guarantee continues to hold in our setting.

**Theorem 7** (Theorem 2 in (Dou et al., 2024)). *Let $\hat{s}$ be given as in Section 2.1. Then*

$$E \left( \int_{-\infty}^{\infty} |\hat{s}(x,t) - s(x,t)|^2 p(x,t)\, dx \right) \lesssim \frac{1}{nt^2}$$

*for $t \geq 1$.*

The following error bound for the plug-in estimator $\widehat{\mathbb{F}}_t := \int_{-\infty}^{\infty} |\hat{s}(x,t) - s_0(x,t)|^2 \, p_0(x,t)\, dx$ defined in Section 2.1 is easily obtained in light of Theorem 7.

**Proposition 5.** *If $t \geq 1$, then*

$$E \left( \left| \widehat{\mathbb{F}}_t - \mathbb{F}_t(f \,||\, f_0) \right| \right) \lesssim \frac{1}{nt^2} + \sqrt{\frac{\mathbb{F}_t(f \,||\, f_0)}{nt^2}}$$

*where $\widehat{\mathbb{F}}_t$ is given in Section 2.1.*

*Proof.* For notational ease, let us write $\mathbb{F}_t$ for $\mathbb{F}_t(f \,||\, f_0)$. Consider

$$|\widehat{\mathbb{F}}_t - \mathbb{F}_t| \lesssim \int_{-\infty}^{\infty} |\hat{s}(x,t) - s(x,t)|^2 p_0(x,t)\, dx + \int_{-\infty}^{\infty} |s(x,t) - s_0(x,t)||\hat{s}(x,t) - s(x,t)|\, p_0(x,t)\, dx$$

$$\lesssim \int_{-\infty}^{\infty} |\hat{s}(x,t) - s(x,t)|^2 p_0(x,t)\, dx + \sqrt{\mathbb{F}_t} \cdot \sqrt{\int_{-\infty}^{\infty} |\hat{s}(x,t) - s(x,t)|^2 \, p_0(x,t)\, dx}.$$

We have used $p(x, t) \asymp p_0(x, t)$ since $f, f_0$ are both bounded above and below by universal constants on their support, and we have also used Cauchy-Schwarz. Therefore, Theorem 7 yields

$$E\left(\left|\widehat{\mathbb{F}}_t - \mathbb{F}_t\right|\right) \lesssim \frac{1}{nt^2} + \sqrt{\frac{\mathbb{F}_t}{nt^2}},$$

as claimed. $\qquad\square$

With the error bound of Proposition 5 in hand, the proof Theorem 1 is straightforward.

*Proof of Theorem 1.* Fix $\eta \in (0, 1)$. Examining the Type I error, consider that since $\mathbb{F}_t(f \,||\, f_0) = 0$ under the null hypothesis, we have

$$P_{f_0}\{\phi_t = 1\} \leq P_{f_0}\left\{|\widehat{\mathbb{F}}_t - \mathbb{F}_t(f \,||\, f_0)| \geq \frac{C'_\eta}{nt^2}\right\} \leq \frac{E\left(\left|\widehat{\mathbb{F}}_t - \mathbb{F}_t(f \,||\, f_0)\right|\right)}{C'_\eta/(nt^2)}$$

by Markov's inequality. Proposition 5 implies there exists some universal constant $\tilde{C} > 0$ such that $E(|\widehat{\mathbb{F}}_t - \mathbb{F}_t(f \,||\, f_0)|) \leq \frac{\tilde{C}}{nt^2}$, and so it follows from taking $C'_\eta$ sufficiently large depending only on $\eta$ that the Type I error is bounded as $P_{f_0}\{\phi_t = 1\} \leq \frac{\eta}{2}$.

Let us now examine the Type II error. Suppose $f \in \mathcal{F}$ with $\mathbb{F}_t(f \,||\, f_0) \geq C\varepsilon_t^2$. Since $C_\eta > 0$ is sufficiently large and $C > C_\eta$, we have by Markov's inequality and Proposition 5,

$$P_f\{\phi_t = 0\} = P_f\left\{\widehat{\mathbb{F}}_t \leq \frac{C'_\eta}{nt^2}\right\} \leq P_f\left\{\mathbb{F}_t(f \,||\, f_0) - \frac{C'_\eta}{nt^2} \leq \left|\mathbb{F}_t(f \,||\, f_0) - \widehat{\mathbb{F}}_t\right|\right\}$$

$$\leq \frac{E\left(\left|\widehat{\mathbb{F}}_t - \mathbb{F}_t(f \,||\, f_0)\right|\right)}{\mathbb{F}_t(f \,||\, f_0) - \frac{C'_\eta}{nt^2}}$$

$$\leq \frac{\tilde{C}/(nt^2)}{\left(C - C'_\eta\right)^2/(nt^2)} + \frac{\tilde{C}\sqrt{\frac{\mathbb{F}_t(f \,||\, f_0)}{nt^2}}}{\mathbb{F}_t(f \,||\, f_0) - \frac{C'_\eta}{nt^2}}.$$

Since $C > C_\eta$, we can take $C_\eta$ sufficiently large to ensure the first term is bounded by $\frac{\eta}{4}$. Furthermore, we can use the inequality $ab \leq a^2 + b^2$ to argue $\tilde{C}\sqrt{\frac{\mathbb{F}_t(f \,||\, f_0)}{nt^2}} \leq \mathbb{F}_t(f \,||\, f_0) \cdot \frac{\eta}{16} + \frac{16\tilde{C}^2}{\eta nt^2}$. Then since $C_\eta$ is sufficiently large, we have $\frac{\tilde{C}\sqrt{\frac{\mathbb{F}_t(f \,||\, f_0)}{nt^2}}}{\mathbb{F}_t(f \,||\, f_0) - \frac{C'_\eta}{nt^2}} \leq \frac{\mathbb{F}_t(f \,||\, f_0) \cdot \frac{\eta}{16}}{\frac{1}{2}\mathbb{F}_t(f \,||\, f_0)} + \frac{\frac{16\tilde{C}^2}{\eta nt^2}}{\frac{C - C'_\eta}{nt^2}} \leq \frac{\eta}{4}$. Putting together our bounds, the Type II error is bounded by $\frac{\eta}{2}$, and so we have shown the sum of the Type I and Type II errors is bounded by $\eta$, completing the proof. $\qquad\square$

## B.2 HIGH NOISE

First, we prove Proposition 1. We later move on to proving Theorem 2 in Appendix B.2.1.

*Proof of Proposition 1.* It immediately follows from $p_0(x, t) \asymp p(x, t)$ that

$$\mathbb{F}_t(f \,||\, f_0) = \int_{-\infty}^{\infty} |s(x, t) - s_0(x, t)|^2 \, p(x, t) \, dx$$

$$= \int_{-\infty}^{\infty} \frac{|\partial_x p(x, t) p_0(x, t) - \partial_x p_0(x, t) p(x, t)|^2}{p_0(x, t)^2 p(x, t)^2} p(x, t) \, dx$$

$$\asymp \int_{-\infty}^{\infty} \frac{|\partial_x p(x, t) p_0(x, t) - \partial_x p_0(x, t) p(x, t)|^2}{p_0(x, t)^3} \, dx$$

$$\leq \int_{-\infty}^{\infty} \frac{|\partial_x p_0(x, t)|^2}{p_0(x, t)^3} |p(x, t) - p_0(x, t)|^2 \, dx + \int_{-\infty}^{\infty} \frac{|\partial_x p(x, t) - \partial_x p_0(x, t)|^2}{p_0(x, t)} \, dx,$$

as claimed. $\qquad\square$

### B.2.1 PROOF OF THEOREM 2

The proof of Theorem 2 will follow once estimation error bounds of $\hat{Q}_t$ and $\hat{Q}'_t$ given by (16) and (17) are obtained. The arguments follow the typical analyses of $U$-statistics, though the calculations are somewhat involved. All the attention is paid to the variances since $\hat{Q}_t$ and $\hat{Q}'_t$ are unbiased for $Q_t$ and $Q'_t$ respectively.

**Proposition 6.** *If $t < 1$, then*

$$E\left(|\hat{Q}_t - Q_t|^2\right) \lesssim \frac{1}{n^2 t^{5/2}} + \frac{Q_t}{nt}$$

*where $\hat{Q}_t$ and $Q_t$ are given by (16) and (14) respectively.*

*Proof of Proposition 6.* Since $\hat{Q}_t$ is unbiased for $Q_t$, it suffices to bound its variance. For notational ease, denote $A(x,t) = \frac{|\partial_x p_0(x,t)|^2}{p_0(x,t)^3}$. By direct calculation, we have

$\text{Var}(\hat{Q}_t)$

$$= \frac{1}{\binom{n}{2}^2} \sum_{i \neq j} \sum_{k \neq l} \int_{-\infty}^{\infty} \int_{-\infty}^{\infty} A(x,t)A(y,t) \cdot$$

$$\text{Cov}\left((\varphi_t(x - \mu_i) - p_0(x,t))(\varphi_t(x - \mu_j) - p_0(x,t)), (\varphi_t(y - \mu_k) - p_0(y,t))(\varphi_t(y - \mu_l) - p_0(y,t))\right) dx\,dy.$$

If $\{i,j\} \cap \{k,l\} = \emptyset$, then the covariance is zero due to independence. Therefore the only cases to consider are when the intersection is nonempty. There are $O(n^2)$ choices of the indices such that $i \neq j, k \neq l$, and $|\{i,j\} \cap \{k,l\}| = 2$. Likewise, there are $O(n^3)$ choices for which $i \neq j, k \neq l$, and $|\{i,j\} \cap \{k,l\}| = 1$. Let $N_1$ and $N_2$ denote the respective counts of choices. Therefore, by the identical distribution of the $\mu_i$'s, we have

$\text{Var}(\hat{Q}_t)$

$$= \frac{N_1}{\binom{n}{2}^2} \int_{-\infty}^{\infty} \int_{-\infty}^{\infty} A(x,t)A(y,t) \cdot$$

$$\text{Cov}((\varphi_t(x - \mu_1) - p_0(x,t))(\varphi_t(x - \mu_2) - p_0(x,t)), (\varphi_t(y - \mu_1) - p_0(x,t))(\varphi_t(y - \mu_2) - p_0(x,t))) dx\,dy$$

$$+ \frac{N_2}{\binom{n}{2}^2} \int_{-\infty}^{\infty} \int_{-\infty}^{\infty} A(x,t)A(y,t) \cdot$$

$$\text{Cov}((\varphi_t(x - \mu_1) - p_0(x,t))(\varphi_t(x - \mu_2) - p_0(x,t)), (\varphi_t(y - \mu_1) - p_0(x,t))(\varphi_t(y - \mu_3) - p_0(x,t))) dx\,dy.$$

By Lemmas 1 and 2, we have $\text{Var}(\hat{Q}_t) \lesssim \frac{1}{n^2 t^{5/2}} + \frac{Q_t}{nt}$, which completes the proof. □

The following lemmas were used in the variance calculation in Proposition 6. Their proofs are deferred to Appendix D.1.

**Lemma 1.** *If $t < 1$, then*

$$\int_{-\infty}^{\infty} \int_{-\infty}^{\infty} A(x,t)A(y,t) \cdot$$

$$\text{Cov}((\varphi_t(x - \mu_1) - p_0(x,t))(\varphi_t(x - \mu_2) - p_0(x,t)), (\varphi_t(y - \mu_1) - p_0(y,t))(\varphi_t(y - \mu_2) - p_0(y,t))) dx\,dy$$

$$\lesssim \frac{1}{t^{5/2}},$$

*where $A(x,t) = \frac{|\partial_x p_0(x,t)|^2}{p_0(x,t)^3}$.*

**Lemma 2.** *If $t < 1$, then*

$$\int_{-\infty}^{\infty} \int_{-\infty}^{\infty} A(x,t)A(y,t) \cdot$$

$$\text{Cov}((\varphi_t(x - \mu_1) - p_0(x,t))(\varphi_t(x - \mu_2) - p_0(x,t)), (\varphi_t(y - \mu_1) - p_0(y,t))(\varphi_t(y - \mu_3) - p_0(y,t))) dx\,dy$$

$$\lesssim \frac{Q_t}{t},$$

*where $A(x,t) = \frac{|\partial_x p_0(x,t)|^2}{p_0(x,t)^3}$ and $Q_t$ is given by (14).*

Proposition 7 gives a similar bound as Proposition 7 for the estimation of $Q'_t$

**Proposition 7.** *If $t < 1$, then*

$$E\left(|\hat{Q}'_t - Q'_t|^2\right) \lesssim \frac{1}{n^2 t^{5/2}} + \frac{Q'_t}{nt}$$

*where $\hat{Q}'_t$ and $Q'_t$ are given by (15) and (17) respectively.*

*Proof of Proposition 7.* Since $\hat{Q}'_t$ is unbiased for $Q'_t$, it suffices to bound its variance. Denote $B(x,t) = \frac{1}{p_0(x,t)}$. Following the same logic as in the proof of Proposition 7, we have

$\mathrm{Var}(\hat{Q}'_t)$

$= \dfrac{N_1}{\binom{n}{2}^2} \displaystyle\int_{-\infty}^{\infty} \int_{-\infty}^{\infty} B(x,t)B(y,t)\cdot$

$\mathrm{Cov}\left((\varphi'_t(x-\mu_1) - \partial_x p_0(x,t))(\varphi'_t(x-\mu_2) - \partial_x p_0(x,t)), (\varphi'_t(y-\mu_1) - \partial_x p_0(y,t))(\varphi'_t(y-\mu_2) - \partial_x p_0(y,t))\right) dx\, dy$

$+ \dfrac{N_2}{\binom{n}{2}^2} \displaystyle\int_{-\infty}^{\infty} \int_{-\infty}^{\infty} B(x,t)B(y,t)\cdot$

$\mathrm{Cov}\left((\varphi'_t(x-\mu_1) - \partial_x p_0(x,t))(\varphi'_t(x-\mu_2) - \partial_x p_0(x,t)), (\varphi'_t(y-\mu_1) - \partial_x p_0(y,t))(\varphi'_t(y-\mu_3) - \partial_x p_0(y,t))\right) dx\, dy.$

By Lemmas 3 and 4, we have $\mathrm{Var}(\hat{Q}'_t) \lesssim \frac{1}{n^2 t^{5/2}} + \frac{Q'_t}{nt}$, which completes the proof. $\qquad\square$

The following lemmas are used in the variance calculation in the argument of Proposition 7, and their proofs are deferred to Appendix D.2.

**Lemma 3.** *If $t < 1$, then*

$\displaystyle\int_{-\infty}^{\infty} \int_{-\infty}^{\infty} B(x,t)B(y,t)\cdot$

$\mathrm{Cov}\left((\varphi'_t(x-\mu_1) - \partial_x p_0(x,t))(\varphi'_t(x-\mu_2) - \partial_x p_0(x,t)), (\varphi'_t(y-\mu_1) - \partial_x p_0(y,t))(\varphi'_t(y-\mu_2) - \partial_x p_0(y,t))\right) dx\, dy$

$\lesssim \dfrac{1}{nt^{5/2}}$

*where $B(x,t) = \frac{1}{p_0(x,t)}$.*

**Lemma 4.** *If $t < 1$, then*

$\displaystyle\int_{-\infty}^{\infty} \int_{-\infty}^{\infty} B(x,t)B(y,t)\cdot$

$\mathrm{Cov}\left((\varphi'_t(x-\mu_1) - \partial_x p_0(x,t))(\varphi'_t(x-\mu_2) - \partial_x p_0(x,t)), (\varphi'_t(y-\mu_1) - \partial_x p_0(y,t))(\varphi'_t(y-\mu_2) - \partial_x p_0(y,t))\right) dx\, dy$

$\lesssim \dfrac{Q'_t}{t}$

*where $B(x,t) = \frac{1}{p_0(x,t)}$ and $Q'_t$ is given by (15).*

Propositions 6 and 7 enable us to prove Theorem 2.

*Proof of Theorem 2.* Fix $\eta \in (0,1)$. By Propositions 6 and 7 along with the inequality $(a+b)^2 \leq 2a^2 + 2b^2$, we have

$$E\left(\left|\hat{Q}_t + \hat{Q}'_t - Q_t - Q'_t\right|^2\right) \leq \tilde{C}\left(\frac{1}{n^2 t^{5/2}} + \frac{Q_t + Q'_t}{nt}\right)$$

for some universal constant $\tilde{C} > 0$. Let us examine the Type I error. Consider by Markov's inequality (and noting $Q_t + Q'_t = 0$ under the null hypothesis),

$$P_{f_0}\{\phi_t = 1\} = P_{f_0}\left\{|\hat{Q}_t + \hat{Q}'_t - Q_t - Q'_t| \geq \frac{C'_\eta}{nt^{5/4}}\right\} \leq \frac{\tilde{C}/(n^2 t^{5/2})}{C'^2_\eta/(n^2 t^{5/2})} = \frac{\tilde{C}}{C'^2_\eta}.$$

By taking $C'_\eta$ sufficiently large depending only on $\eta$, it follows the Type I error is bounded by $\eta/2$.

We now turn to the Type II error. Suppose $f \in \mathcal{F}$ with $\mathbb{F}_t(f \| f_0) \geq C\varepsilon_t^2$. By Proposition 1, there exists some universal constant $\tilde{C}' > 0$ such that $C\varepsilon_t^2 \leq \tilde{C}'(Q_t + Q'_t)$. Since $C_\eta > 0$ is sufficiently large, we have by Markov's inequality

$$
\begin{aligned}
P_f\{\phi_t = 0\} &\leq P\left\{|\hat{Q}_t + \hat{Q}'_t| \leq \frac{C'_\eta}{nt^{5/4}}\right\} \\
&\leq P\left\{Q_t + Q'_t - \frac{C'_\eta}{nt^{5/4}} \leq |\hat{Q}_t + \hat{Q}'_t - Q_t - Q'_t|\right\} \\
&\leq \frac{E\left(|\hat{Q}_t + \hat{Q}'_t - Q_t - Q'_t|^2\right)}{\left(Q_t + Q'_t - \frac{C'_\eta}{nt^{5/4}}\right)^2} \\
&\leq \frac{\tilde{C}\left(\frac{1}{n^2 t^{5/2}} + \frac{Q_t + Q'_t}{nt}\right)}{\left(Q_t + Q'_t - \frac{C'_\eta}{nt^{5/4}}\right)^2} \\
&\leq \frac{\frac{\tilde{C}}{n^2 t^{5/2}} + \frac{16\tilde{C}^2}{\eta} \cdot \frac{1}{n^2 t^2} + \frac{\eta}{16}(Q_t + Q'_t)^2}{\left(Q_t + Q'_t - \frac{C'_\eta}{nt^{5/4}}\right)^2} \\
&\leq \frac{\frac{\tilde{C}}{n^2 t^{5/2}} + \frac{16\tilde{C}^2}{\eta} \cdot \frac{1}{n^2 t^2}}{(C/2)^2 \cdot \frac{1}{n^2 t^{5/2}}} + \frac{\frac{\eta}{16}(Q_t + Q'_t)^2}{((Q_t + Q'_t)/2)^2} \\
&\leq \frac{\eta}{4} + \frac{\eta}{4} \\
&\leq \frac{\eta}{2}.
\end{aligned}
$$

Here, we have used the inequality $ab \leq a^2 + b^2$ in the above calculation. We have also used $\frac{C}{\tilde{C}'}\varepsilon_t^2 \leq Q_t + Q'_t$ and $C > 0$ is sufficiently large in the above calculation. Hence, the Type II error is bounded by $\frac{\eta}{2}$, and so the sum of the Type I and Type II errors is at most $\eta$. The proof is complete. $\square$

### B.3 Low noise: Proof of Theorem 4

Like in Appendix B.2.1, the proof of Theorem 4 rests on estimation error results for $\hat{Q}_K$ and $\hat{Q}'_K$ given by (19) and (20) respectively. These orthogonal series estimators are classical (Laurent, 1996); we provide full proof details of Lemmas 5 and 6 in Appendix E.2 for completeness though the ideas are not new.

**Lemma 5.** *We have*

$$
E\left(|\hat{Q}_K - Q|^2\right) \lesssim K^{-4\alpha} + \frac{K}{n^2} + \frac{Q}{n}
$$

*where $Q$ is defined in Proposition 3 and $\hat{Q}_K$ is given by (19).*

**Lemma 6.** *If $\alpha \geq 1$, we have*

$$
E\left(|\hat{Q}'_K - Q'|^2\right) \lesssim K^{-4(\alpha-1)} + \frac{K^5}{n^2} + \frac{K^2 Q'}{n}
$$

*where $Q'$ is defined in Proposition 3 and $\hat{Q}'_K$ is given by (20).*

*Proof of Theorem 4.* Fix $\eta \in (0,1)$. By Lemmas 5 and 6, along with the inequality $(a+b)^2 \leq 2a^2 + 2b^2$ and the choice of $K \asymp n^{\frac{2}{4\alpha+1}}$, we have

$$
E\left(\left|\hat{Q}_K + \hat{Q}'_K - Q - Q'\right|^2\right) \leq \tilde{C}\left(n^{-\frac{8(\alpha-1)}{4\alpha+1}} + \frac{n^{\frac{4}{4\alpha+1}}(Q + Q')}{n}\right)
$$

for some universal constant $\tilde{C} > 0$. Examining the Type I error, consider by Markov's inequality (and noting $Q + Q' = 0$ under the null hypothesis),

$$P_{f_0} \{\phi = 1\} = P_{f_0} \left\{ |\hat{Q}_K + \hat{Q}'_K - Q - Q'| \geq C'_\eta n^{-\frac{4(\alpha-1)}{4\alpha+1}} \right\} \leq \frac{\tilde{C} n^{-\frac{8(\alpha-1)}{4\alpha+1}}}{C'^2_\eta n^{-\frac{8(\alpha-1)}{4\alpha+1}}} \leq \frac{\tilde{C}}{C'^2_\eta} \leq \frac{\eta}{2}$$

where the final inequality follows by taking $C'_\eta > 0$ sufficiently large. Hence, the Type I error is bounded by $\eta/2$.

Let us now examine the Type II error. Suppose $f \in \mathcal{F}_\alpha$ with $\mathbb{F}_t(f \,||\, f_0) \geq C\varepsilon_t^2$. By Proposition 3, there exists some universal constant $\tilde{C}' > 0$ such that

$$Cn^{-\frac{4(\alpha-1)}{4\alpha+1}} = C\varepsilon_t^2 \leq \tilde{C}' \left( Q + Q' + t^{\alpha-1} \right) \leq \tilde{C}' \left( Q + Q' + n^{-\frac{4(\alpha-1)}{4\alpha+1}} \right).$$

Since $C_\eta > 0$ can be taken sufficiently large, it follows that $\frac{C}{2} n^{-\frac{4(\alpha-1)}{4\alpha+1}} \leq \tilde{C}(Q + Q')$. Then since $C_\eta > 0$ is sufficiently large, we have by Markov's inequality

$$
\begin{aligned}
P_f \{\phi = 0\} &= P \left\{ |\hat{Q}_K + \hat{Q}'_K| \leq C'_\eta n^{-\frac{4(\alpha-1)}{4\alpha+1}} \right\} \\
&\leq P \left\{ Q + Q' - C'_\eta n^{-\frac{4(\alpha-1)}{4\alpha+1}} \leq |\hat{Q}_K + \hat{Q}'_K - Q - Q'| \right\} \\
&\leq \frac{E \left( |\hat{Q}_K + \hat{Q}'_K - Q - Q'|^2 \right)}{\left( Q + Q' - C'_\eta n^{-\frac{4(\alpha-1)}{4\alpha+1}} \right)^2} \\
&\leq \frac{\tilde{C} \left( n^{-\frac{8(\alpha-1)}{4\alpha+1}} + \frac{n^{\frac{4}{4\alpha+1}}(Q+Q')}{n} \right)}{\left( Q + Q' - C'_\eta n^{-\frac{4(\alpha-1)}{4\alpha+1}} \right)^2} \\
&\leq \frac{\tilde{C} \left( n^{-\frac{8(\alpha-1)}{4\alpha+1}} + \frac{16\tilde{C}}{\eta} \frac{n^{\frac{8}{4\alpha+1}}}{n^2} + \frac{\eta}{16\tilde{C}}(Q+Q')^2 \right)}{\left( Q + Q' - C'_\eta n^{-\frac{4(\alpha-1)}{4\alpha+1}} \right)^2} \\
&\leq \frac{\tilde{C} \left( n^{-\frac{8(\alpha-1)}{4\alpha+1}} + \frac{16\tilde{C}}{\eta} n^{-\frac{8(\alpha-1)}{4\alpha+1} - \frac{4}{4\alpha+1}} \right)}{(C/4)^2 n^{-\frac{8(\alpha-1)}{4\alpha+1}}} + \frac{\frac{\eta}{16}(Q+Q')^2}{((Q+Q')/2)^2} \\
&\leq \frac{\tilde{C} + \frac{16\tilde{C}^2}{\eta}}{(C/4)^2} + \frac{\eta}{4} \\
&\leq \frac{\eta}{2}
\end{aligned}
$$

where the final inequality follows from taking $C_\eta > 0$ sufficiently large and noting $C > C_\eta$. Therefore, the Type II error is bounded by $\frac{\eta}{2}$, and so the sum of Type I and Type II errors is bounded above by $\eta$. The proof is complete. $\qquad\square$

### B.4 Aggregation: testing in TV distance

In this section, we prove Theorem 6. To reason about the aggregated test statistic $\int_0^T \widehat{\mathbb{U}}_t \, dt$ with $\widehat{\mathbb{U}}_t$ given by (22), it is useful to think about the estimand that $\widehat{\mathbb{U}}_t$ targets. Define

$$\mathbb{U}_t = \begin{cases} \mathbb{F}_t(f \,||\, f_0) & \text{if } t \geq 1, \\ Q_t + Q'_t & \text{if } n^{-\frac{4}{4\alpha+1}} < t < 1, \\ Q + Q' & \text{if } t \leq n^{-\frac{4}{4\alpha+1}}, \end{cases} \tag{24}$$

where $Q_t$ and $Q'_t$ are given by (14) and (15) respectively, and $Q$ and $Q'$ are defined in Proposition 3. Proposition 8 gives a bound on the estimation error of the aggregated statistic $\int_0^T \widehat{\mathbb{U}}_t \, dt$.

**Proposition 8.** *Suppose $T > 0$. If $c > 0$, then there exists $C > 0$ such that*

$$E\left(\left|\int_0^T \widehat{\mathbb{U}}_t \, dt - \int_0^T \mathbb{U}_t \, dt\right|\right) \le c \int_0^T \mathbb{U}_t \, dt + Cn^{-\frac{4\alpha}{4\alpha+1}},$$

*where $\widehat{\mathbb{U}}_t$ and $\mathbb{U}_t$ are given by (22) and (24) respectively.*

*Proof.* Observe

$$E\left(\left|\int_0^T \widehat{\mathbb{U}}_t \, dt - \int_0^T \mathbb{U}_t \, dt\right|\right) \le \int_0^\infty E\left(\left|\widehat{\mathbb{U}}_t - \mathbb{U}_t\right|\right) \, dt.$$

We will split the integral into the three regions $t \le n^{-\frac{4}{4\alpha+1}}, n^{-\frac{4}{4\alpha+1}} < t < 1$, and $t \ge 1$. Consider by Proposition 5, we have

$$\int_1^\infty E\left(\left|\widehat{\mathbb{U}}_t - \mathbb{U}_t\right|\right) \, dt = \int_1^\infty E\left(\left|\widehat{\mathbb{F}}_t - \mathbb{F}_t(f \,\|\, f_0)\right|\right) \, dt$$

$$\lesssim \int_1^\infty \frac{1}{nt^2} + \sqrt{\frac{\mathbb{F}_t}{nt^2}} \, dt$$

$$\le \frac{C}{n} + c \int_1^\infty \mathbb{F}_t \, dt$$

$$= \frac{C}{n} + c \int_1^\infty \mathbb{U}_t \, dt,$$

where $C > 0$ is a sufficiently large quantity depending only on $c > 0$. Here, we have used the inequality $ab \le \frac{a^2}{r^2} + r^2 b^2$ for any $r > 0$.

Consider over the region $n^{-\frac{4}{4\alpha+1}} < t < 1$, we have from Propositions 6 and 7 as well as a very similar argument,

$$\int_{n^{-\frac{4}{4\alpha+1}}}^1 E\left(\left|\widehat{\mathbb{U}}_t - \mathbb{U}_t\right|\right) \, dt \le \int_{n^{-\frac{4}{4\alpha+1}}}^1 E\left(\left|\hat{Q}_t - Q_t\right|\right) + E\left(\left|\hat{Q}'_t - Q_t - Q'_t\right|\right) \, dt$$

$$\lesssim \int_{n^{-\frac{4}{4\alpha+1}}}^1 \frac{1}{nt^{5/4}} + \sqrt{\frac{Q_t}{nt}} + \sqrt{\frac{Q'_t}{nt}} \, dt$$

$$\le Cn^{-\frac{4\alpha}{4\alpha+1}} + c \int_{n^{-\frac{4}{4\alpha+1}}}^1 \mathbb{U}_t \, dt.$$

A very similar argument using Lemmas 5 and 6 yields, over the region $t \le n^{-\frac{4}{4\alpha+1}}$, the bound

$$\int_0^{n^{-\frac{4}{4\alpha+1}}} E\left(\left|\widehat{\mathbb{U}}_t - \mathbb{U}_t\right|\right) \, dt \le C \int_0^{n^{-\frac{4}{4\alpha+1}}} n^{-\frac{4(\alpha-1)}{4\alpha+1}} \, dt + c \int_0^{n^{-\frac{4}{4\alpha+1}}} \mathbb{U}_t \, dt$$

$$= Cn^{-\frac{4\alpha}{4\alpha+1}} + c \int_0^{n^{-\frac{4}{4\alpha+1}}} \mathbb{U}_t \, dt.$$

Putting together our three bounds over the three regions yields the desired result. $\qquad\square$

Proposition 8 gives the key error bound in arguing for the success of aggregation for testing (5). Let us describe the intuition here. Under the alternative hypothesis $d_{\text{TV}}(f, f_0)^2 \ge \varepsilon^2$ with $\varepsilon^2 \gtrsim n^{-\frac{4\alpha}{4\alpha+1}}$, we have from (6) that $\varepsilon^2 \lesssim \int_0^T \mathbb{F}_t(f \,\|\, f_0) \, dt \lesssim \int_0^T \mathbb{U}_t \, dt$. In other words, there is signal to detect. Proposition 8 implies

$$\int_0^T \widehat{\mathbb{U}}_t \, dt \ge (1 - c) \int_0^T \mathbb{U}_t(f \,\|\, f_0) \, dt - Cn^{-\frac{4\alpha}{4\alpha+1}}$$

with high probability under the alternative. Similarly, under the null hypothesis where $\mathbb{U}_t(f \,\|\, f_0) = 0$, we have $\int_0^T \widehat{\mathbb{U}}_t \, dt \le Cn^{-\frac{4\alpha}{4\alpha+1}}$ with high probability. Consequently, it follows we can detect

the alternative when $\int_0^T \mathbb{U}_t(f \,||\, f_0)\, dt \geq \varepsilon^2 \geq C'n^{-\frac{4\alpha}{4\alpha+1}}$ with $C' > 0$ being a sufficiently large constant. Hence, the minimax rate $(\varepsilon^*)^2 \asymp n^{-\frac{4\alpha}{4\alpha+1}}$ is achieved, notably without any extraneous logarithmic terms. The proof of Theorem 6 implements this intuition.

*Proof of Theorem 6.* Fix $\eta \in (0,1)$. Examining the Type I error, consider that under the null hypothesis of (5), we have $\mathbb{U}_t = 0$ for all $t > 0$. Therefore, by Markov's inequality we have

$$
\begin{aligned}
P_{f_0}\{\phi_T = 1\} &= P_{f_0}\left\{\int_0^T \widehat{\mathbb{U}}_t\, dt \geq C'_\eta n^{-\frac{4\alpha}{4\alpha+1}}\right\} \\
&\leq P_{f_0}\left\{\left|\int_0^T \widehat{\mathbb{U}}_t\, dt - \int_0^T \mathbb{U}_t\, dt\right| \geq C'_\eta n^{-\frac{4\alpha}{4\alpha+1}}\right\} \\
&\leq \frac{E\left(\left|\int_0^T \widehat{\mathbb{U}}_t\, dt - \int_0^T \mathbb{U}_t\, dt\right|\right)}{C'_\eta n^{-\frac{4\alpha}{4\alpha+1}}} \\
&\leq \frac{\tilde{C}n^{-\frac{4\alpha}{4\alpha+1}}}{C'_\eta n^{-\frac{4\alpha}{4\alpha+1}}} \\
&\leq \frac{\tilde{C}}{C'_\eta}
\end{aligned}
$$

where we have applied Proposition 8, which implies there exists some universal constant $\tilde{C} > 0$ such that $E\left(\left|\int_0^T \widehat{\mathbb{U}}_t\, dt - \int_0^T \mathbb{U}_t\, dt\right|\right) \leq \tilde{C}n^{-\frac{4\alpha}{4\alpha+1}}$ since $\mathbb{U}_t = 0$ under the null hypothesis. Therefore, it follows from taking $C'_\eta$ sufficiently large depending only on $\eta$ that the Type I error is bounded as $P_{f_0}\{\phi_t = 1\} \leq \frac{\eta}{2}$.

Let us now examine the Type II error. Suppose $f \in \mathcal{F}_\alpha$ with $\mathrm{d}_{\mathrm{TV}}(f, f_0) \geq Cn^{-\frac{2\alpha}{4\alpha+1}}$. Now, since $T \gtrsim n$, it follows from the fact $f$ and $f_0$ are compactly supported that $\mathrm{d}_{\mathrm{KL}}(f * \varphi_T \,||\, f_0 * \varphi_T) \lesssim \frac{1}{T} \lesssim \frac{1}{n}$. Therefore, it follows by (6) and by Propositions 1 and 3 that

$$
\begin{aligned}
C^2 n^{-\frac{4\alpha}{4\alpha+1}} \leq \mathrm{d}_{\mathrm{TV}}(f, f_0)^2 &\lesssim \frac{1}{n} + \int_0^T \mathbb{F}_t(f \,||\, f_0)\, dt \\
&\lesssim \frac{1}{n} + \int_0^T \mathbb{U}_t\, dt + \int_0^{n^{-\frac{4}{4\alpha+1}}} t^{\alpha-1}\, dt \\
&\asymp n^{-\frac{4\alpha}{4\alpha+1}} + \int_0^T \mathbb{U}_t\, dt.
\end{aligned}
$$

Therefore, by taking $C > C_\eta > 0$ sufficiently large, it follows that we must have

$$
\int_0^T \mathbb{U}_t\, dt \geq Cn^{-\frac{4\alpha}{4\alpha+1}}.
$$

With such a bound in hand, along with Proposition 8, we can employ very similar arguments as those appearing in the proofs of Theorems 1, 2, and 4 to bound the Type II error by $\frac{\eta}{2}$, provided $C_\eta > 0$ is chosen sufficiently large, thus yielding the claimed result. We omit the details. $\qquad\square$

## C  LOWER BOUNDS

In this section, we present the minimax lower bounds which were deferred from the main text.

### C.1  TESTING IN FISHER DIVERGENCE

For the problem (4) under the parameter space $\mathcal{F}$ given by (9), Theorem 8 establishes the minimax lower bound $\frac{1}{nt^{5/4}} \wedge \frac{1}{t}$ for $t \lesssim 1$.

**Theorem 8.** *There exists a universal constant $\tilde{c} > 0$ such that the following holds. If $t \leq \tilde{c}$ and $\eta \in (0, 1)$, then there exists $c_\eta > 0$ depending only on $\eta$ such that for all $0 < c < c_\eta$, we have*

$$\inf_\phi \left\{ P_{f_0}\{\phi = 1\} + \sup_{\substack{f \in \mathcal{F}, \\ \mathbb{F}_t(f \,||\, f_0) \geq c\varepsilon_t^2}} P_f\{\phi = 0\} \right\} \geq 1 - \eta,$$

*with $\varepsilon_t^2 = \frac{1}{nt^{5/4}} \wedge \frac{1}{t}$.*

Before we prove Theorem 8, we first make some high level remarks. Our argument starts by constructing a prior distribution on the composite alternative $\{f \in \mathcal{F} : \mathbb{F}_t(f \,||\, f_0) \geq c\varepsilon_t^2\}$, and works to show it is difficult to distinguish between the null hypothesis $P_{f_0}$ and the resulting mixture distribution induced by our choice of prior by bounding the $\chi^2$-divergence. The details of the construction build on recent developments on lower bound arguments for score estimation (Dou et al., 2024). Our prior will be supported on the collection $\{f_b\}_{b \in \{-1,1\}^m}$ with

$$f_b(\mu) = f_0(\mu) + \lambda \sum_{i=1}^m b_i \omega \left( \frac{\mu - x_i}{\rho} \right) \tag{25}$$

where $\lambda$ and $\rho$ are parameters to be chosen, $\{x_i\}_{i=1}^m$ are grid points in $[-1, 1]$ which are spaced $2\rho$ apart with $m \asymp \frac{1}{\rho}$, and $\omega : \mathbb{R} \to \mathbb{R}$ is a function supported on $[-1, 1]$ such that $\omega \in C^\infty(\mathbb{R})$, $\int_{-\infty}^\infty \omega(x)\, dx = 0$, and $\|w\|_\infty \lesssim 1$. Finally, we make the choice $\rho \asymp \sqrt{t}$ and $\lambda \asymp \frac{1}{\sqrt{n}t^{1/8}} \wedge 1$. The choice of $\lambda$ and $\rho$ are different in our setting since Dou et al. (2024) deal with Hölder smooth densities whereas $\mathcal{F}$ imposes no smoothness assumptions.

For $t \gtrsim 1$, the minimax rate (12) specializes to $\frac{1}{nt^2}$. Theorem 7 in (Dou et al., 2024) directly establishes this lower bound, and so we omit the proof. Their use of Le Cam's two-point method in the score estimation problem can also be applied to our testing problem. It is not surprising that the result from the estimation problem can be employed here, since $t \gtrsim 1$ is essentially the parametric regime in which the limits of testing and estimation coincide.

We now dive in to proving Theorem 8. It is easy to verify that $f_b \in \mathcal{F}$ provided $\lambda$ is smaller than a sufficiently small universal constant. Hence, Proposition 12 is stated without proof.

**Proposition 9.** *If $\lambda \leq c$ where $c > 0$ is a sufficiently small universal constant, then $\{f_b\}_{b \in \{-1,1\}^m} \subset \mathcal{F}$.*

We use the uniform prior on $\{f_b\}_{b \in \{-1,1\}^m}$ in the lower bound argument. In terms of the data, the Bayes testing problem can be expressed as

$$
\begin{aligned}
H_0 &: (\mu_1, ..., \mu_n) \sim f_0^{\otimes n}, \\
H_1 &: (\mu_1, ..., \mu_n) \sim \frac{1}{2^m} \sum_{b \in \{-1,1\}^m} f_b^{\otimes n}.
\end{aligned}
\tag{26}
$$

It needs to be checked that the Bayes problem (29) is a valid reduction of (4). In order to do so, the separation between any $f_b$ and the null hypothesis $f_0$ needs to be computed. Quantitative bounds on the separation are available from Dou et al. (2024) and is one of the major contributions of their paper.

**Proposition 10** ((Dou et al., 2024)). *There exist universal constants $C, c_1, c_2 > 0$ such that if $t \leq c_1$, $\rho \leq c_2$, and $\rho = C\sqrt{t}$, then*

$$\mathbb{F}_t(f_b \,||\, f_0) \gtrsim \frac{\lambda^2 m}{\rho}$$

*for $b \in \{-1, 1\}^m$. Here, $f_b$ is given by (25).*

*Proof.* In (Dou et al., 2024), see (60) in the proof of Theorem 6 and the use of Proposition 4 in the argument to derive (60). Note that the reasoning of Dou et al. (2024) applies by the correspondence $\lambda = \epsilon^\alpha$ with $\alpha > 0$ arbitrarily small. $\qquad\square$

**Proposition 11.** *There exists some universal constant $C > 0$ such that*

$$\chi^2 \left( \frac{1}{2^m} \sum_{b \in \{-1,1\}^m} f_b^{\otimes n} \;\middle\|\; f_0^{\otimes n} \right) \leq e^{Cn^2\lambda^4\rho^2 m} - 1,$$

*where $f_b$ is given by (25).*

*Proof.* For notational ease, write $f_b = f_0 + \psi_b$. Note $\int_{-1}^{1} \psi_b(\mu)\, d\mu = 0$. Direct calculation yields

$$1 + \chi^2 \left( \frac{1}{2^m} \sum_{b \in \{-1,1\}^m} f_b^{\otimes n} \;\middle\|\; f_0^{\otimes n} \right)$$

$$= \int_{[-1,1]^n} \frac{\left( 2^{-m} \sum_{b \in \{-1,1\}^m} f_b^{\otimes n}(\mu) \right)^2}{f_0^{\otimes n}(\mu)} \, d\mu$$

$$= 2^{-2m} \sum_{b,b' \in \{-1,1\}^m} \int_{[-1,1]^n} \frac{f_b^{\otimes n}(\mu) f_{b'}^{\otimes n}(\mu)}{f_0^{\otimes n}(\mu)} \, d\mu$$

$$= 2^{-2m} \sum_{b,b' \in \{-1,1\}^m} \prod_{i=1}^{n} \int_{-1}^{1} \frac{f_b(\mu_i) f_{b'}(\mu_i)}{f_0(\mu_i)} \, d\mu_i$$

$$= 2^{-2m} \sum_{b,b' \in \{-1,1\}^m} \left( 1 + \int_{-1}^{1} \frac{\psi_b(\mu)\psi_{b'}(\mu)}{f_0(\mu)} \, d\mu \right)^n$$

$$= 2^{-2m} \sum_{b,b' \in \{-1,1\}^m} \left( 1 + \lambda^2 \sum_{i,j=1}^{m} b_i b_j' \int_{-1}^{1} \frac{\omega\left(\frac{\mu - x_i}{\rho}\right) \omega\left(\frac{\mu - x_j}{\rho}\right)}{f_0(\mu)} \, d\mu \right)^n$$

$$= 2^{-2m} \sum_{b,b' \in \{-1,1\}^m} \left( 1 + \lambda^2 \sum_{i=1}^{m} b_i b_i' \int_{-1}^{1} \frac{\omega\left(\frac{\mu - x_i}{\rho}\right)^2}{f_0(\mu)} \, d\mu \right)^n.$$

The final line follows from the fact that $\omega\left(\frac{\cdot - x_i}{\rho}\right)$ and $\omega\left(\frac{\cdot - x_j}{\rho}\right)$ have disjoint supports for $i \neq j$.

Denote $w_i := \int_{-1}^{1} \frac{\omega\left(\frac{\mu - x_i}{\rho}\right)^2}{f_0(\mu)} \, d\mu$. Continuing with the calculation, consider

$$2^{-2m} \sum_{b,b' \in \{-1,1\}^m} \left( 1 + \lambda^2 \sum_{i=1}^{m} b_i b_i' w_i \right)^n \leq 2^{-2m} \sum_{b,b' \in \{-1,1\}^m} \exp\left( n\lambda^2 \sum_{i=1}^{m} b_i b_i' w_i \right)$$

$$= E\left( \exp\left( n\lambda^2 \sum_{i=1}^{m} R_i w_i \right) \right),$$

where $R_i \overset{iid}{\sim} \text{Rademacher}(1/2)$. By independence, we have

$$E\left( \exp\left( n\lambda^2 \sum_{i=1}^{m} R_i w_i \right) \right) = \prod_{i=1}^{n} E\left( \exp\left( n\lambda^2 R_i w_i \right) \right) = \prod_{i=1}^{m} \cosh\left( n\lambda^2 w_i \right) \leq \exp\left( n^2 \lambda^4 \sum_{i=1}^{m} w_i^2 \right).$$

Since $f_0(\mu) \asymp 1$ for $|\mu| \leq 1$, it follows that $w_i \asymp \int_{-1}^{1} \omega\left(\frac{\mu - x_i}{\rho}\right)^2 d\mu \asymp \rho$. Therefore, $\exp\left( n^2\lambda^4 \sum_{i=1}^{m} w_i^2 \right) \leq \exp\left( Cn^2\lambda^4 m\rho^2 \right)$ for some universal constant $C > 0$. $\qquad\square$

We are now in position to prove Theorem 8. Roughly speaking, $\lambda$ and $\rho$ are to be tuned subject to the constraint $\lambda \leq c$ for a sufficiently small universal constant, and such that the $\chi^2$ divergence between $2^{-m} \sum_{b \in \{-1,1\}^m} f_b^{\otimes n}$ and $f_0^{\otimes n}$ can be bounded by an arbitrarily small constant. Proposition 10 already specifies the choice $\rho \asymp \sqrt{t}$, and so it remains to select $\lambda$.

*Proof of Theorem 8.* Fix $\eta \in (0, 1)$. By Propositions 12 and 13, there exist some universal constants $\tilde{C}, \tilde{c} > 0$ such that for $t \leq \tilde{c}$, $\lambda \leq \tilde{c}$, and $\rho = \tilde{C}\sqrt{t}$, we have

$$\mathbb{F}_t(f_b \,\|\, f_0) \gtrsim \frac{\lambda^2}{t} \tag{27}$$

for all $b \in \{-1, 1\}^m$, where $f_b$ is given by (25). Here, we have used $m \asymp \frac{1}{\rho}$. By Proposition 14, there exists some universal constant $C > 0$ such that

$$\chi^2 \left( \frac{1}{2^m} \sum_{b \in \{-1,1\}^m} f_b^{\otimes n} \,\middle\|\, f_0^{\otimes n} \right) \leq e^{Cn^2 \lambda^4 \sqrt{t}} - 1$$

where, again, we have used $m \asymp \frac{1}{\rho}$ and $\rho \asymp \sqrt{t}$. Select

$$\lambda = \left( \frac{\eta^2}{Ce} \right)^{1/4} \left( \frac{1}{\sqrt{n} t^{1/8}} \wedge 1 \right).$$

It is clear $Cn^2 \lambda^4 \sqrt{t} \leq \frac{\eta^2}{e} < 1$. From the inequality $e^x - 1 \leq ex$ for $x \in (0, 1)$, we thus have

$$\mathrm{d}_{\mathrm{TV}} \left( \frac{1}{2^m} \sum_{b \in \{-1,1\}^m} f_b^{\otimes n}, f_0^{\otimes n} \right) \leq \frac{1}{2} \sqrt{\chi^2 \left( \frac{1}{2^m} \sum_{b \in \{-1,1\}^m} f_b^{\otimes n} \,\middle\|\, f_0^{\otimes n} \right)} \leq \frac{1}{2} \sqrt{\eta^2} \leq \eta. \tag{28}$$

With our choice of $\lambda$ and (30), consider

$$\mathbb{F}_t(f_b \,\|\, f_0) \geq \kappa_\eta \left( \frac{1}{nt^{5/4}} \wedge \frac{1}{t} \right)$$

for some $\kappa_\eta > 0$ depending only on $\eta$. Take $c_\eta = \kappa_\eta$ and observe that $\{f_b\}_{b \in \{-1,1\}^m} \subset \{f \in \mathcal{F} : \mathbb{F}_t(f \,\|\, f_0) \geq c\varepsilon_t^2\}$ for all $c < c_\eta$. Therefore, it follows

$$\inf_\phi \left\{ P_{f_0} \{\phi = 1\} + \sup_{\substack{f \in \mathcal{F}, \\ \mathbb{F}_t(f \| f_0) \geq c\varepsilon_t^2}} P_f \{\phi = 0\} \right\} \geq \inf_\phi \left\{ P_{f_0} \{\phi = 1\} + \sup_{b\{-1,1\}^m} P_{f_b} \{\phi = 0\} \right\}$$

$$\geq \inf_\phi \left\{ P_{f_0} \{\phi = 1\} + \frac{1}{2^m} \sum_{b \in \{-1,1\}^m} P_{f_b} \{\phi = 0\} \right\}$$

$$= 1 - \mathrm{d}_{\mathrm{TV}} \left( \frac{1}{2^m} \sum_{b \in \{-1,1\}^m} f_b^{\otimes n}, f_0^{\otimes n} \right)$$

$$\geq 1 - \eta$$

where the penultimate line follows from Neyman-Pearson lemma and the final line follows from (31). The proof is complete. $\qquad\square$

## C.2    Testing in Fisher divergence with smoothness

Theorem 9 establishes the minimax lower bound $\frac{1}{nt^{5/4}} \wedge \left( n^{-\frac{4(\alpha-1)}{4\alpha+1}} + t^{\alpha-1} \right)$, showing that the improved rate achieved by the projection estimator in Section 3.1 is optimal.

**Theorem 9.** *There exists a universal constant $\tilde{c} > 0$ such that the following holds. If $t \leq \tilde{c}$ and $\eta \in (0, 1)$, then there exists $c_\eta > 0$ depending only on $\eta$ such that for all $0 < c < c_\eta$, we have*

$$\inf_\phi \left\{ P_{f_0} \{\phi = 1\} + \sup_{\substack{f \in \mathcal{F}_\alpha, \\ \mathbb{F}_t(f \| f_0) \geq c\varepsilon_t^2}} P_f \{\phi = 0\} \right\} \geq 1 - \eta,$$

*with $\varepsilon_t^2 = \frac{1}{nt^{5/4}} \wedge \left( n^{-\frac{4(\alpha-1)}{4\alpha+1}} + t^{\alpha-1} \right)$.*

The argument is very similar to the proof of Theorem 8. However, some modification is needed to ensure the constructed densities live in $\mathcal{F}_\alpha$ (instead of just $\mathcal{F}$). Our prior will be supported on the collection $\{f_b\}_{b \in \{-1,1\}^m}$ with

$$f_b(\mu) = f_0(\mu) + \epsilon^\alpha \sum_{i=1}^m b_i \omega \left( \frac{\mu - x_i}{\rho} \right).$$

Here, $0 < \epsilon \leq \rho < 1$ are parameters to be chosen, and the constraint $\epsilon \leq \rho$ is important to ensure the Hölder smoothness. However, we will also require that the first $\lfloor \alpha \rfloor$ derivatives of $\omega$ are bounded. Our prior is the uniform distribution on $\{f_b\}_{b \in \{-1,1\}^m}$. Finally, we make the choice $\rho \asymp \sqrt{t} \vee \epsilon$ and $\epsilon \asymp \left( \frac{1}{n^2 \sqrt{t}} \right)^{1/(4\alpha)} \wedge \left( n^{-\frac{2}{4\alpha+1}} \mathbb{1}_{\{\alpha \geq 1\}} + \sqrt{t} \mathbb{1}_{\{\alpha < 1\}} \right)$. The forms of the densities are precisely the forms employed by Dou et al. (2024), especially more so now since we also enforce $\epsilon \leq \rho$ as they do. We remark that the choice of $\epsilon$ is different from the choice made in (Dou et al., 2024); this difference is expected since the hypothesis testing problem we consider has different minimax rates than the score estimation problem.

The proof of Theorem 9 follows the same general path as the proof of Theorem 8. First, it is easy to verify that $f_b \in \mathcal{F}_\alpha$ provided $\epsilon \leq \rho$ is smaller than a sufficiently small universal constant. Moreover, note that $f_b$ satisfies the periodicity constraint since $\omega$ is compactly supported. Hence, Proposition 12 is stated without proof.

**Proposition 12.** *If $\epsilon \leq \rho \leq c$ where $c > 0$ is a sufficiently small universal constant, then $\{f_b\}_{b \in \{-1,1\}^m} \subset \mathcal{F}_\alpha$.*

Recall we use the uniform prior on $\{f_b\}_{b \in \{-1,1\}^m}$, and the Bayes testing problem is

$$
\begin{aligned}
H_0 &: (\mu_1, ..., \mu_n) \sim f_0^{\otimes n}, \\
H_1 &: (\mu_1, ..., \mu_n) \sim \frac{1}{2^m} \sum_{b \in \{-1,1\}^m} f_b^{\otimes n}.
\end{aligned}
\tag{29}
$$

Proposition 13 is an analogue of Proposition 10, but note that the choice of $\rho$ satisfies the constraint $\rho \geq \epsilon$ to ensure the Hölder condition is satisfied.

**Proposition 13** ((Dou et al., 2024))**.** *There exist universal constants $C, c_1, c_2 > 0$ such that if $t \leq c_1$, $\epsilon \leq c_2$, and $\rho = C\sqrt{t} \vee \epsilon$, then*

$$\mathbb{F}_t(f_b \,\|\, f_0) \gtrsim \frac{\epsilon^{2\alpha} m}{\rho}$$

*for $b \in \{-1, 1\}^m$.*

*Proof.* In (Dou et al., 2024), see (60) in the proof of Theorem 6 and the use of Proposition 4 in the argument to derive (60). $\square$

**Proposition 14.** *There exists some universal constant $C > 0$ such that*

$$\chi^2 \left( \frac{1}{2^m} \sum_{b \in \{-1,1\}^m} f_b^{\otimes n} \,\Big\|\, f_0^{\otimes n} \right) \leq e^{Cn^2 \epsilon^{4\alpha} \rho^2 m} - 1.$$

*Proof.* The proof is exactly the same as Proposition 11 with $\epsilon^\alpha = \lambda$. $\square$

We can now prove Theorem 9. We tune $\rho$ and $\epsilon$ subject to the constraint $\rho \geq \epsilon$ and $\epsilon \leq c$ for a sufficiently small universal constant, and such that the $\chi^2$ divergence between the null and alternative hypotheses can be bounded by an arbitrarily small constant. Proposition 13 already specifies the choice $\rho \asymp \sqrt{t} \vee \epsilon$, and so it remains to select $\epsilon$.

*Proof of Theorem 9.* Fix $\eta \in (0, 1)$. By Propositions 12 and 13, there exist some universal constants $\tilde{C}, \tilde{c} > 0$ such that for $t \leq \tilde{c}$, $\epsilon \leq \tilde{c}$, and $\rho = \tilde{C}\sqrt{t} \vee \epsilon$, we have

$$\mathbb{F}_t(f_b \,\|\, f_0) \gtrsim \frac{\epsilon^{2\alpha}}{t} \wedge \epsilon^{2(\alpha-1)} \tag{30}$$

for all $b \in \{-1,1\}^m$. Here, we have used $m \asymp \frac{1}{\rho}$. By Proposition 14, there exists some universal constant $C > 0$ such that

$$\chi^2 \left( \frac{1}{2^m} \sum_{b \in \{-1,1\}^m} f_b^{\otimes n} \,\middle\|\, f_0^{\otimes n} \right) \leq e^{Cn^2 \epsilon^{4\alpha}(\sqrt{t} \vee \epsilon)} - 1$$

where, again, we have used $m \asymp \frac{1}{\rho}$ and $\rho \asymp \sqrt{t} \vee \epsilon$. Select

$$\epsilon = \left( \left( \frac{\eta^2}{Ce} \right)^{\frac{1}{4\alpha}} \left( \frac{1}{n^2\sqrt{t}} \right)^{\frac{1}{4\alpha}} \right) \wedge \left( \left( \frac{\eta^2}{Ce} \right)^{\frac{1}{4\alpha+1}} \left( n^{-\frac{2}{4\alpha+1}} \cdot \mathbb{1}_{\{\alpha \geq 1\}} + \sqrt{t} \cdot \mathbb{1}_{\{\alpha < 1\}} \right) \right).$$

We claim $Cn^2 \epsilon^{4\alpha}(\sqrt{t} \vee \epsilon) \leq \frac{\eta^2}{e} < 1$. To see this, consider it suffices to show both $Cn^2 \epsilon^{4\alpha} \sqrt{t} \leq \frac{\eta^2}{Ce}$ and $Cn^2 \epsilon^{4\alpha+1} \leq \frac{\eta^2}{Ce}$ hold. The first condition is easily seen to be satisfied since $\epsilon \leq \left( \frac{\eta^2}{Ce} \right)^{\frac{1}{4\alpha}} \left( \frac{1}{n^2\sqrt{t}} \right)^{\frac{1}{4\alpha}}$. To show the second condition, suppose $\alpha \geq 1$. Then $\epsilon \leq \left( \frac{\eta^2}{Ce} \right)^{\frac{1}{4\alpha+1}} n^{-\frac{2}{4\alpha+1}}$, which immediately establishes the second condition holds. Suppose $\alpha < 1$. Since we have both $\epsilon \leq \left( \frac{\eta^2}{Ce} \right)^{\frac{1}{4\alpha}} \left( \frac{1}{n^2\sqrt{t}} \right)^{\frac{1}{4\alpha}}$ and $\epsilon \leq \left( \frac{\eta^2}{Ce} \right)^{\frac{1}{4\alpha+1}} \sqrt{t}$, it follows that $\epsilon^{4\alpha+1} \leq \frac{\eta^2}{Ce} \cdot \frac{1}{n^2\sqrt{t}} \cdot \left( \frac{\eta^2}{Ce} \right)^{\frac{1}{4\alpha+1}} \sqrt{t}$, which delivers $Cn^2 \epsilon^{4\alpha+1} \leq \frac{\eta^2}{Ce} \cdot \left( \frac{\eta^2}{Ce} \right)^{\frac{1}{4\alpha+1}} \leq \frac{\eta^2}{e}$ since we can assume $C \geq 1$ without loss of generality. Hence, we have shown the claim.

From the inequality $e^x - 1 \leq ex$ for $x \in (0,1)$, it follows from $Cn^2 \epsilon^{4\alpha}(\sqrt{t} \vee \epsilon) \leq \frac{\eta^2}{e} < 1$ that we have

$$\mathrm{d}_{\mathrm{TV}} \left( \frac{1}{2^m} \sum_{b \in \{-1,1\}^m} f_b^{\otimes n}, f_0^{\otimes n} \right) \leq \frac{1}{2} \sqrt{\chi^2 \left( \frac{1}{2^m} \sum_{b \in \{-1,1\}^m} f_b^{\otimes n} \,\middle\|\, f_0^{\otimes n} \right)} \leq \frac{1}{2}\sqrt{\eta^2} \leq \eta. \quad (31)$$

With our choice of $\epsilon$ and (30), consider

$$\mathbb{F}_t(f_b \,\|\, f_0) \geq \kappa_\eta \left( \frac{1}{nt^{5/4}} \wedge \left( n^{-\frac{4(\alpha-1)}{4\alpha+1}} + t^{\alpha-1} \right) \right)$$

for some $\kappa_\eta > 0$ depending only on $\eta$. Take $c_\eta = \kappa_\eta$ and observe that $\{f_b\}_{b \in \{-1,1\}^m} \subset \{f \in \mathcal{F}_\alpha : \mathbb{F}_t(f \,\|\, f_0) \geq c\varepsilon_t^2\}$ for all $c < c_\eta$. Therefore, it follows

$$\inf_\phi \left\{ P_{f_0}\{\phi = 1\} + \sup_{\substack{f \in \mathcal{F}_\alpha, \\ \mathbb{F}_t(f \,\|\, f_0) \geq c\varepsilon_t^2}} P_f\{\phi = 0\} \right\} \geq \inf_\phi \left\{ P_{f_0}\{\phi = 1\} + \sup_{b\{-1,1\}^m} P_{f_b}\{\phi = 0\} \right\}$$

$$\geq \inf_\phi \left\{ P_{f_0}\{\phi = 1\} + \frac{1}{2^m} \sum_{b \in \{-1,1\}^m} P_{f_b}\{\phi = 0\} \right\}$$

$$= 1 - \mathrm{d}_{\mathrm{TV}} \left( \frac{1}{2^m} \sum_{b \in \{-1,1\}^m} f_b^{\otimes n}, f_0^{\otimes n} \right)$$

$$\geq 1 - \eta$$

where the penultimate line follows from Neyman-Pearson lemma and the final line follows from (31). The proof is complete. $\qquad\square$

## D    HIGH NOISE: DEFERRED PROOFS

This section contains the deferred proofs of Lemmas 1, 2, 3, and 4.

### D.1 Proofs of Lemmas 1 and 2

*Proof of Lemma 1.* We proceed by direct calculation. Consider the covariance satisfies

$$\text{Cov}\left((\varphi_t(x-\mu_1)-p_0(x,t))(\varphi_t(x-\mu_2)-p_0(x,t)),(\varphi_t(y-\mu_1)-p_0(y,t))(\varphi_t(y-\mu_2)-p_0(y,t))\right)$$

$$= \left(E\left((\varphi_t(x-\mu_1)-p_0(x,t))(\varphi_t(y-\mu_1)-p_0(y,t))\right)\right)^2 - \left(E\left(\varphi_t(x-\mu_1)-p_0(x,t)\right)\right)^2\left(E\left(\varphi_t(y-\mu_1)-p_0(y,t)\right)\right)^2$$

$$= \int_{-1}^{1}\int_{-1}^{1}(\varphi_t(x-\mu)-p_0(x,t))(\varphi_t(y-\mu)-p_0(y,t))(\varphi_t(x-\nu)-p_0(x,t))(\varphi_t(y-\nu)-p_0(y,t))\,f(\mu)f(\nu)\,d\mu\,d\nu$$

$$- (p(x,t)-p_0(x,t))^2(p(y,t)-p_0(y,t))^2.$$

Therefore,

$$\int_{-\infty}^{\infty}\int_{-\infty}^{\infty}A(x,t)A(y,t)\cdot$$
$$\text{Cov}\left((\varphi_t(x-\mu_1)-p_0(x,t))(\varphi_t(x-\mu_2)-p_0(x,t)),(\varphi_t(y-\mu_1)-p_0(y,t))(\varphi_t(y-\mu_2)-p_0(y,t))\right)\,dx\,dy$$

$$= \int_{-1}^{1}\int_{-1}^{1}\left(\int_{-\infty}^{\infty}A(x,t)(\varphi_t(x-\mu)-p_0(x,t))(\varphi_t(x-\nu)-p_0(x,t))\,dx\right)^2 f(\mu)f(\nu)\,d\mu\,d\nu$$

$$- \int_{-\infty}^{\infty}\int_{-\infty}^{\infty}A(x,t)A(y,t)(p(x,t)-p_0(x,t))^2(p(y,t)-p_0(y,t))^2\,dx\,dy$$

$$= \int_{-1}^{1}\int_{-1}^{1}\left(\int_{-\infty}^{\infty}A(x,t)(\varphi_t(x-\mu)-p_0(x,t))(\varphi_t(x-\nu)-p_0(x,t))\,dx\right)^2 f(\mu)f(\nu)\,d\mu\,d\nu - Q_t^2$$

$$\lesssim \int_{-1}^{1}\int_{-1}^{1}\left(\int_{-\infty}^{\infty}A(x,t)(\varphi_t(x-\mu)-p_0(x,t))(\varphi_t(x-\nu)-p_0(x,t))\,dx\right)^2 d\mu\,d\nu$$

where we have used $f_0(\mu) \asymp 1$ for $\mu \in [-1,1]$. We can split into three terms,

$$\int_{-1}^{1}\int_{-1}^{1}\left(\int_{-\infty}^{\infty}A(x,t)(\varphi_t(x-\mu)-p_0(x,t))(\varphi_t(x-\nu)-p_0(x,t))\,dx\right)^2 d\mu\,d\nu$$

$$\lesssim \int_{-1}^{1}\int_{-1}^{1}\left(\int_{-\infty}^{\infty}A(x,t)\varphi_t(x-\mu)\varphi_t(x-\nu)\,dx\right)^2 d\mu\,d\nu \tag{32}$$

$$+ \int_{-1}^{1}\left(\int_{-\infty}^{\infty}A(x,t)\varphi_t(x-\mu)p_0(x,t)\,dx\right)^2 d\mu \tag{33}$$

$$+ \left(\int_{-\infty}^{\infty}A(x,t)p_0(x,t)^2\,dx\right)^2. \tag{34}$$

The terms (33) and (34) can be handled similarly. Consider that by Jensen's inequality, we have the bound $\int_{-1}^{1}\left(\int_{-\infty}^{\infty}A(x,t)\varphi_t(x-\mu)p_0(x,t)\,dx\right)^2 d\mu \leq \int_{-1}^{1}\int_{-\infty}^{\infty}A(x,t)^2 p_0(x,t)^2\varphi_t(x-\mu)\,dx\,d\mu \asymp \int_{-\infty}^{\infty}A(x,t)^2 p_0(x,t)^3\,dx$ since $f_0(\mu) \asymp 1$ for $\mu \in [-1,1]$. Similarly, looking at (34), we have by Jensen's inequality $\left(\int_{-\infty}^{\infty}A(x,t)p_0(x,t)^2\,dx\right)^2 = \left(\int_{-\infty}^{\infty}A(x,t)p_0(x,t)\cdot p_0(x,t)\,dx\right)^2 \leq \int_{-\infty}^{\infty}A(x,t)^2 p_0(x,t)^3\,dx$, and so (33) and (34) both admit the same bound. From here, observe

$$\int_{-\infty}^{\infty}A(x,t)^2 p_0(x,t)^3\,dx = \int_{-\infty}^{\infty}\frac{|\partial_x p_0(x,t)|^4}{p_0(x,t)^3}\,dx.$$

Consider by Holder's inequality with the choice $p = 4$ and $q = \frac{4}{3}$, we have

$$
\int_{-\infty}^{\infty} \frac{|\partial_x p_0(x,t)|^4}{p_0(x,t)^3}\, dx = \int_{-\infty}^{\infty} \frac{\left| \int_{-1}^{1} \frac{x-\mu}{t} \varphi_t(x-\mu) f_0(\mu)\, d\mu \right|^4}{p_0(x,t)^3}\, dx
$$

$$
\leq \int_{-\infty}^{\infty} \frac{\left( \int_{-1}^{1} \frac{|x-\mu|^4}{t^4} \varphi_t(x-\mu)\, d\mu \right) \left( \int_{-1}^{1} \varphi_t(x-\mu)\, d\mu \right)^3}{p_0(x,t)^3}
$$

$$
\asymp \int_{-1}^{1} \int_{-\infty}^{\infty} \frac{|x-\mu|^4}{t^4} \varphi_t(x-\mu)\, d\mu\, dx
$$

$$
\lesssim \frac{1}{t^2}.
$$

Therefore, we have shown that both (33) and (34) are of order at most $\frac{1}{t^2}$.

It remains to bound (32). Let us write $h_\nu(x) = A(x,t)\varphi_t(x-\nu)$. Then

$$
(32) = \int_{-1}^{1} \int_{-1}^{1} \left( \int_{-\infty}^{\infty} h_\nu(x)\varphi_t(x-\mu)\, dx \right)^2 d\mu\, d\nu
$$

$$
= \int_{-1}^{1} \int_{-1}^{1} |(h_\nu * \varphi_t)(\mu)|^2\, d\mu\, d\nu
$$

$$
\leq \int_{-1}^{1} \int_{-1}^{1} |h_\nu(\mu)|^2\, d\mu\, d\nu
$$

$$
= \int_{-1}^{1} \int_{-1}^{1} |A(\mu,t)\varphi_t(\mu-\nu)|^2\, d\mu\, d\nu
$$

$$
= \int_{-1}^{1} \int_{-1}^{1} \frac{|s_0(\mu,t)|^4 \varphi_t(\mu-\nu)^2}{p_0(\mu,t)^2}\, d\mu\, d\nu.
$$

Here, we have used that $\|g * \varphi_t\|_{L_2(\mathbb{R})}^2 \leq \|g\|_{L_2(\mathbb{R})}^2$ for any function $g : \mathbb{R} \to \mathbb{R}$. Continuing with the calculation, consider $p_0(\mu,t) \asymp 1$ for $\mu \in [-1,1]$. Furthermore, we have $\varphi_t(\mu-\nu)^2 \lesssim \frac{1}{t^{1/2}} \varphi_t(\mu-\nu)$. Additionally, from our earlier calculation, it is straightforward to see $|s_0(\mu,t)|^4 \asymp |\partial_x p_0(\mu,t)|^4 \lesssim \frac{1}{t^2}$. Hence, we have

$$
\int_{-1}^{1} \int_{-1}^{1} \frac{|s_0(\mu,t)|^4 \varphi_t(\mu-\nu)^2}{p_0(\mu,t)^2}\, d\mu\, d\nu \lesssim \int_{-1}^{1} \int_{-1}^{1} \frac{1}{t^{5/2}} \varphi_t(\mu-\nu)\, d\mu\, d\nu \lesssim \frac{1}{t^{5/2}}.
$$

Putting together this bound with the bound $\frac{1}{t^2}$ for (33) and (34) shown earlier, it follows that

$$
\int_{-1}^{1} \int_{-1}^{1} \left( \int_{-\infty}^{\infty} A(x,t)(\varphi_t(x-\mu) - p_0(x,t))(\varphi_t(x-\nu) - p_0(x,t))\, dx \right)^2 d\mu\, d\nu \lesssim \frac{1}{t^{5/2}},
$$

completing the proof. $\square$

*Proof of Lemma 2.* The covariance satisfies

$\text{Cov}((\varphi_t(x-\mu_1) - p_0(x,t))(\varphi_t(x-\mu_2) - p_0(x,t)), (\varphi_t(y-\mu_1) - p_0(x,t))(\varphi_t(y-\mu_3) - p_0(y,t)))$

$= E((\varphi_t(x-\mu_1) - p_0(x,t))(\varphi_t(y-\mu_1) - p_0(y,t)))E(\varphi_t(x-\mu_1) - p_0(x,t))E(\varphi_t(y-\mu_1) - p_0(y,t))$

$- (E(\varphi_t(x-\mu_1) - p_0(x,t)))^2(E(\varphi_t(y-\mu_1) - p_0(y,t)))^2$

$= E((\varphi_t(x-\mu_1) - p_0(x,t))(\varphi_t(y-\mu_1) - p_0(y,t)))(p(x,t) - p_0(x,t))(p(y,t) - p_0(y,t))$

$- (p(x,t) - p_0(x,t))^2(p(y,t) - p_0(y,t))^2.$

Here, we have used that $\mu_i$ and $\mu_j$ for $i \neq j$ are independent and identically distributed to obtain the second line. With this in hand, it follows that

$$\int_{-\infty}^{\infty} \int_{-\infty}^{\infty} A(x,t)A(y,t)\cdot$$

$$\text{Cov}((\varphi_t(x - \mu_1) - p_0(x,t))(\varphi_t(x - \mu_2) - p_0(x,t)), (\varphi_t(y - \mu_1) - p_0(x,t))(\varphi_t(y - \mu_3) - p_0(x,t))) \, dx \, dy$$

$$= \int_{-\infty}^{\infty} \int_{-\infty}^{\infty} A(x,t)A(y,t)\cdot$$

$$E((\varphi_t(x - \mu_1) - p_0(x,t))(\varphi_t(y - \mu_1) - p_0(y,t)))(p(x,t) - p_0(x,t))(p(y,t) - p_0(y,t)) \, dx \, dy$$

$$- \int_{-\infty}^{\infty} \int_{-\infty}^{\infty} A(x,t)A(y,t)(p(x,t) - p_0(x,t))^2(p(y,t) - p_0(y,t))^2 \, dx \, dy$$

$$= \int_{-1}^{1} \left( \int_{-\infty}^{\infty} A(x,t)(\varphi_t(x - \mu) - p_0(x,t))(p(x,t) - p_0(x,t)) \, dx \right)^2 f(\mu) \, d\mu - Q_t^2$$

$$\leq \int_{-1}^{1} \left( \int_{-\infty}^{\infty} |A(x,t)||\varphi_t(x - \mu) - p_0(x,t)|^2 \, dx \right) \left( \int_{-\infty}^{\infty} |A(x,t)||p(x,t) - p_0(x,t)|^2 \, dx \right) f(\mu) \, d\mu$$

$$= Q_t \int_{-1}^{1} \int_{-\infty}^{\infty} A(x,t)|\varphi_t(x - \mu) - p_0(x,t)|^2 f(\mu) \, d\mu \, dx$$

$$\lesssim Q_t \left( \int_{-\infty}^{\infty} A(x,t) \int_{-1}^{1} \varphi_t(x - \mu)^2 f(\mu) \, d\mu \, dx + \int_{-\infty}^{\infty} A(x,t)p_0(x,t)^2 \, dx \right).$$

Here, we have applied Cauchy-Schwarz to obtain the third-to-last line in the previous display. Note, it is important $A$ is nonnegative so that $|A| = A$, yielding $Q_t = \int_{-\infty}^{\infty} A(x,t)|p(x,t) - p_0(x,t)|^2 \, dx$ and delivering the penultimate line. By definition of $A$, we have $\int_{-\infty}^{\infty} A(x,t)p_0(x,t)^2 \, dx = \int_{-\infty}^{\infty} |s_0(x,t)|^2 p_0(x,t) \, dx \lesssim \frac{1}{t}$. Likewise, observe from Lemma 7 that

$$\int_{-\infty}^{\infty} A(x,t) \int_{-1}^{1} \varphi_t(x - \mu)^2 f(\mu) \, d\mu \, dx$$

$$\lesssim \int_{-\infty}^{\infty} A(x,t) \left( \frac{1}{\sqrt{t}} \mathbb{1}_{\{|x| \leq 1\}} + \frac{1}{t} e^{-\frac{(|x|-1)^2}{t}} \mathbb{1}_{\{|x| > 1\}} \right) \, dx$$

$$= \frac{1}{\sqrt{t}} \int_{|x| \leq 1} |\partial_x p_0(x,t)|^2 \, dx + \frac{1}{t} \int_{|x| > 1} \left( \frac{|\partial_x p_0(x,t)|^2}{p_0(x,t)^3} \right) e^{-\frac{(|x|-1)^2}{t}} \, dx$$

$$\lesssim \frac{1}{t} + \frac{1}{t} \int_{|x| > 1} \left( \frac{|x| - 1}{\sqrt{t}} \vee 1 \right)^3 \cdot \frac{1}{\sqrt{t}} e^{-\frac{(|x|-1)^2}{2t}} \, dx$$

$$\lesssim \frac{1}{t}.$$

Here, we have used integration by parts and Assumption 1 to conclude that $|\partial_x p_0(x,t)|^2 = \left| \int_{-1}^{1} \frac{x-\mu}{t} \varphi_t(x - \mu) f_0(\mu) \, d\mu \right|^2 = \left| f_0(-1)\varphi_t(x + 1) - f_0(1)\varphi_t(x - 1) + \int_{-1}^{1} f_0'(\mu)\varphi_t(x - \mu) \, d\mu \right|^2 \lesssim \varphi_t(x + 1)^2 + \varphi_t(x - 1)^2 + 1$ since $||f_0'||_\infty \lesssim 1$. This gives us $\int_{|x| \leq 1} |\partial_x p_0(x,t)|^2 \, dx \lesssim \frac{1}{\sqrt{t}}$, which we used to obtain the penultmiate line. Hence, we have shown the bound $\frac{Q_t}{t}$ as claimed. $\square$

## D.2 Proofs of Lemmas 3 and 4

*Proof of Lemma 3.* The proof broadly follows the same structure as the proof of Lemma 1. Consider the covariance satisfies

$$\text{Cov}\left((\varphi_t'(x - \mu_1) - \partial_x p_0(x,t))(\varphi_t'(x - \mu_2) - \partial_x p_0(x,t)), (\varphi_t'(y - \mu_1) - \partial_x p_0(y,t))(\varphi_t'(y - \mu_2) - \partial_x p_0(y,t))\right)$$

$$= \left(E\left((\varphi_t'(x - \mu_1) - \partial_x p_0(x,t))(\varphi_t'(y - \mu_1) - \partial_x p_0(y,t))\right)\right)^2 - (\partial_x p(x,t) - \partial_x p_0(x,t))^2(\partial_x p(y,t) - \partial_x p_0(y,t))^2.$$

Therefore,

$$\int_{-\infty}^{\infty} \int_{-\infty}^{\infty} B(x,t)B(y,t)\cdot$$

$$\text{Cov}\left((\varphi_t'(x-\mu_1) - \partial_x p_0(x,t))(\varphi_t'(x-\mu_2) - \partial_x p_0(x,t)), (\varphi_t'(y-\mu_1) - \partial_x p_0(y,t))(\varphi_t'(y-\mu_2) - \partial_x p_0(y,t))\right) \, dx \, dy$$

$$= \int_{-\infty}^{\infty} \int_{-\infty}^{\infty} \int_{-1}^{1} \int_{-1}^{1} B(x,t)B(y,t)(\varphi_t'(x-\mu) - \partial_x p_0(x,t))(\varphi_t'(y-\mu) - \partial_x p_0(y,t))\cdot$$

$$(\varphi_t'(x-\nu) - \partial_x p_0(x,t))(\varphi_t'(y-\nu) - \partial_x p_0(y,t)) f(\mu) \, f(\nu) \, d\mu \, d\nu \, dx \, dy$$

$$- \int_{-\infty}^{\infty} \int_{-\infty}^{\infty} B(x,t)(\partial_x p(x,t) - \partial_x p_0(x,t))^2 B(y,t)(\partial_x p(y,t) - \partial_x p_0(y,t))^2 \, dx \, dy$$

$$= \int_{-1}^{1} \int_{-1}^{1} \left( \int_{-\infty}^{\infty} B(x,t)(\varphi_t'(x-\mu) - \partial_x p_0(x,t))(\varphi_t'(x-\nu) - \partial_x p_0(x,t)) \, dx \right)^2 f(\mu)f(\nu) \, d\mu \, d\nu - Q_t'^2$$

$$\lesssim \int_{-1}^{1} \int_{-1}^{1} \left( \int_{-\infty}^{\infty} B(x,t)\varphi_t'(x-\mu)\varphi_t'(x-\nu) \, dx \right)^2 d\mu \, d\nu + \int_{-1}^{1} \left( \int_{-\infty}^{\infty} B(x,t)\varphi_t'(x-\mu)\partial_x p_0(x,t) \, dx \right)^2 d\mu$$

$$+ \left( \int_{-\infty}^{\infty} B(x,t)(\partial_x p_0(x,t))^2 \, dx \right)^2.$$

Looking at the third term and recalling $B(x,t) = \frac{1}{p_0(x,t)}$, it is clear $\left( \int_{-\infty}^{\infty} B(x,t)(\partial_x p_0(x,t))^2 \, dx \right)^2 = \left( \int_{-\infty}^{\infty} s_0(x,t)^2 p_0(x,t) \, dx \right)^2 \lesssim \frac{1}{t^2}$. Similarly, the second term satisfies

$$\int_{-1}^{1} \left( \int_{-\infty}^{\infty} B(x,t)\varphi_t'(x-\mu)\partial_x p_0(x,t) \, dx \right)^2 d\mu$$

$$= \int_{-1}^{1} \left( \int_{-\infty}^{\infty} B(x,t)\frac{-(x-\mu)}{t}\varphi_t(x-\mu)\partial_x p_0(x,t) \, dx \right)^2 d\mu$$

$$\leq \int_{-1}^{1} \left( \int_{-\infty}^{\infty} \frac{(x-\mu)^2}{t^2}\varphi_t(x-\mu) \, dx \right) \left( \int_{-\infty}^{\infty} B(x,t)^2(\partial_x p_0(x,t))^2 \varphi_t(x-\mu) \, dx \right) d\mu$$

$$= \frac{1}{t} \int_{-1}^{1} \int_{-\infty}^{\infty} s_0(x,t)^2 \varphi_t(x-\mu) \, dx \, d\mu$$

$$\lesssim \frac{1}{t^2}$$

since $\int_{-1}^{1} \int_{-\infty}^{\infty} s_0(x,t)^2 \varphi_t(x-\mu) \, dx \, d\mu = \int_{-\infty}^{\infty} s_0(x,t)^2 p_0(x,t) \, dx \, d\mu \lesssim \frac{1}{t}$. It remains to analyze the first term. For notational ease, let us denote $h_\nu(x) = B(x,t)\varphi_t'(x-\nu)$. Observe $h_\nu$ is differentiable everywhere. Since $\varphi_t'(x-\mu) = -\varphi_t'(\mu-x)$, it follows

$$\int_{-1}^{1} \int_{-1}^{1} \left( \int_{-\infty}^{\infty} B(x,t)\varphi_t'(x-\mu)\varphi_t'(x-\nu) \, dx \right)^2 d\mu \, d\nu$$

$$= \int_{-1}^{1} \int_{-1}^{1} |(\varphi_t' * h_\nu)(\mu)|^2 \, d\mu \, d\nu$$

$$= \int_{-1}^{1} \int_{-1}^{1} |(\varphi_t * h_\nu')(\mu)|^2 \, d\mu \, d\nu$$

$$\leq \int_{-1}^{1} \int_{-1}^{1} |h_\nu'(\mu)|^2 \, d\mu \, d\nu$$

$$= \int_{-1}^{1} \int_{-1}^{1} \frac{|\varphi_t''(\mu-\nu)p_0(\mu,t) - \varphi_t'(\mu-\nu)\partial_x p_0(\mu,t)|^2}{p_0(\mu,t)^4} \, d\mu \, d\nu.$$

We have used $B(x, t) = \frac{1}{p_0(x,t)}$ here. By Lemma 7, we have $p_0(\mu, t) \asymp 1$ for $|\mu| \leq 1$. Therefore,

$$\int_{-1}^{1} \int_{-1}^{1} \frac{|\varphi_t''(\mu - \nu)p_0(\mu, t) - \varphi_t'(\mu - \nu)\partial_x p_0(\mu, t)|^2}{p_0(\mu, t)^4} \, d\mu \, d\nu$$

$$\lesssim \int_{-1}^{1} \int_{-1}^{1} |\varphi_t''(\mu - \nu)|^2 \, d\mu \, d\nu + \int_{-1}^{1} \int_{-1}^{1} (\varphi_t'(\mu - \nu)\partial_x p_0(\mu, t))^2 \, d\mu \, d\nu$$

$$= \int_{-1}^{1} \int_{-1}^{1} \left| -\frac{1}{t}\varphi_t(\mu - \nu) + \frac{(\mu - \nu)^2}{t^2}\varphi_t(\mu - \nu) \right|^2 \, d\mu \, d\nu + \int_{-1}^{1} \int_{-1}^{1} \frac{(\mu - \nu)^2}{t^2}\varphi_t(\mu - \nu)^2 |\partial_x p_0(\mu, t)|^2 \, d\mu \, d\nu$$

$$\lesssim \int_{-1}^{1} \int_{-1}^{1} \frac{1}{t^2}\varphi_t(\mu - \nu)^2 + \frac{(\mu - \nu)^4}{t^4}\varphi_t(\mu - \nu)^2 \, d\mu \, d\nu + \int_{-1}^{1} \int_{-1}^{1} \frac{(\mu - \nu)^2}{t^3}\varphi_t(\mu - \nu)^2 \, d\mu \, d\nu$$

$$\lesssim \int_{-1}^{1} \int_{-1}^{1} \frac{1}{t^{5/2}}\varphi_t(\mu - \nu) + \frac{(\mu - \nu)^4}{t^{9/2}}\varphi_t(\mu - \nu) \, d\mu \, d\nu + \frac{1}{t^{5/2}} \int_{-1}^{1} \int_{-1}^{1} \frac{(\mu - \nu)^2}{t}\varphi_t(\mu - \nu) \, d\mu \, d\nu$$

$$\lesssim \frac{1}{t^{5/2}}.$$

In the above calculation, we have used $|\partial_x p_0(\mu, t)| = \left| \int_{-1}^{1} \frac{\mu - \zeta}{t}\varphi_t(\mu - \zeta)f_0(\zeta) \, d\zeta \right| \lesssim \frac{1}{\sqrt{t}}$ and $\varphi_t(\mu - \nu)^2 \lesssim \frac{1}{\sqrt{t}}\varphi_t(\mu - \nu)$. Hence, we have shown the bound

$$\text{Cov}\left( (\varphi_t'(x - \mu_1) - \partial_x p_0(x, t))(\varphi_t'(x - \mu_2) - \partial_x p_0(x, t)), (\varphi_t'(y - \mu_1) - \partial_x p_0(y, t))(\varphi_t'(y - \mu_2) - \partial_x p_0(y, t)) \right)$$

$$= (E\left( (\varphi_t'(x - \mu_1) - \partial_x p_0(x, t))(\varphi_t'(y - \mu_1) - \partial_x p_0(y, t)) \right))^2 - (\partial_x p(x, t) - \partial_x p_0(x, t))^2(\partial_x p(y, t) - \partial_x p_0(y, t))^2$$

$$\lesssim \frac{1}{t^2} + \frac{1}{t^{5/2}}$$

$$\asymp \frac{1}{t^{5/2}}$$

since $t < 1$. The proof is complete. $\qquad \square$

*Proof of Lemma 4.* The proof follows the same cadence as the proof of Lemma 2. The covariance satisfies

$$\text{Cov}\left( (\varphi_t'(x - \mu_1) - \partial_x p_0(x, t))(\varphi_t'(x - \mu_2) - \partial_x p_0(x, t)), (\varphi_t(y - \mu_1) - p_0(y, t))(\varphi_t'(y - \mu_3) - \partial_x p_0(y, t)) \right)$$

$$= E\left( (\varphi_t'(x - \mu_1) - \partial_x p_0(x, t))(\varphi_t'(y - \mu_1) - \partial_x p_0(y, t)) \right) E\left( \varphi_t'(x - \mu_1) - \partial_x p_0(x, t) \right) E(\varphi_t'(y - \mu_1) - \partial_x p_0(y, t))$$

$$- (E\left( \varphi_t'(x - \mu_1) - \partial_x p_0(x, t) \right))^2 (E\left( \varphi_t'(y - \mu_1) - \partial_x p_0(y, t) \right))^2$$

$$= E\left( (\varphi_t'(x - \mu_1) - \partial_x p_0(x, t))(\varphi_t'(y - \mu_1) - \partial_x p_0(y, t)) \right) (\partial_x p(x, t) - \partial_x p_0(x, t))(\partial_x p(y, t) - \partial_x p_0(y, t))$$

$$- (\partial_x p(x, t) - \partial_x p_0(x, t))^2(\partial_x p(y, t) - \partial_x p_0(y, t))^2.$$

Consider

$$\int_{-\infty}^{\infty} \int_{-\infty}^{\infty} B(x, t)B(y, t) \cdot$$

$$\text{Cov}\left( (\varphi_t'(x - \mu_1) - \partial_x p_0(x, t))(\varphi_t'(x - \mu_2) - \partial_x p_0(x, t)), (\varphi_t'(y - \mu_1) - \partial_x p_0(y, t))(\varphi_t'(y - \mu_3) - \partial_x p_0(y, t)) \right) \, dx \, dy$$

$$= \int_{-\infty}^{\infty} \int_{-\infty}^{\infty} B(x, t)B(y, t) \cdot$$

$$E\left( (\varphi_t'(x - \mu_1) - \partial_x p_0(x, t))(\varphi_t'(y - \mu_1) - \partial_x p_0(y, t)) \right) (\partial_x p(x, t) - \partial_x p_0(x, t))(\partial_x p(y, t) - \partial_x p_0(y, t)) \, dx \, dy$$

$$- \int_{-\infty}^{\infty} \int_{-\infty}^{\infty} B(x, t)B(y, t)(\partial_x p(x, t) - \partial_x p_0(x, t))^2(\partial_x p(y, t) - \partial_x p_0(y, t))^2 \, dx \, dy$$

$$= \int_{-1}^{1} \left( \int_{-\infty}^{\infty} B(x, t)(\varphi_t'(x - \mu) - \partial_x p_0(x, t))(\partial_x p(x, t) - \partial_x p_0(x, t)) \, dx \right)^2 f(\mu) \, d\mu - Q_t'^2$$

$$\lesssim \int_{-1}^{1} \left( \int_{-\infty}^{\infty} B(x, t)\varphi_t'(x - \mu)(\partial_x p(x, t) - \partial_x p_0(x, t)) \, dx \right)^2 f(\mu) \, d\mu$$

$$+ \left( \int_{-\infty}^{\infty} B(x, t)\partial_x p_0(x, t)(\partial_x p(x, t) - \partial_x p_0(x, t)) \, dx \right)^2.$$

Recalling $B(x,t) = \frac{1}{p_0(x,t)}$, Cauchy-Schwarz yields

$$\int_{-1}^{1} \left( \int_{-\infty}^{\infty} B(x,t)\varphi_t'(x-\mu)(\partial_x p(x,t) - \partial_x p_0(x,t)) \, dx \right)^2 f(\mu) \, d\mu$$

$$= \int_{-1}^{1} \left( \int_{-\infty}^{\infty} -\frac{x-\mu}{t} \varphi_t(x-\mu) B(x,t)(\partial_x p(x,t) - \partial_x p_0(x,t)) \, dx \right)^2 f(\mu) \, d\mu$$

$$\leq \int_{-1}^{1} \left( \int_{-\infty}^{\infty} \frac{(x-\mu)^2}{t^2} \varphi_t(x-\mu) \, dx \right) \left( \int_{-\infty}^{\infty} \frac{(\partial_x p(x,t) - \partial_x p_0(x,t))^2}{p_0(x,t)^2} \varphi_t(x-\mu) \, dx \right) f(\mu) \, d\mu$$

$$= \frac{1}{t} \int_{-1}^{1} \int_{-\infty}^{\infty} \frac{(\partial_x p(x,t) - \partial_x p_0(x,t))^2}{p_0(x,t)^2} \varphi_t(x-\mu) f(\mu) \, d\mu \, dx$$

$$= \frac{1}{t} \int_{-\infty}^{\infty} \frac{(\partial_x p(x,t) - \partial_x p_0(x,t))^2}{p_0(x,t)^2} p(x,t) \, dx$$

$$\lesssim \frac{1}{t} \int_{-\infty}^{\infty} B(x,t)(\partial_x p(x,t) - \partial_x p_0(x,t))^2 \, dx$$

$$= \frac{Q_t'}{t}.$$

Here, we have used $p(x,t) \asymp p_0(x,t)$ to obtain the penultimate line. Likewise, observe Cauchy-Schwarz gives

$$\left( \int_{-\infty}^{\infty} B(x,t)\partial_x p_0(x,t)(\partial_x p(x,t) - \partial_x p_0(x,t)) \, dx \right)^2$$

$$= \left( \int_{-\infty}^{\infty} \frac{(\partial_x p(x,t) - \partial_x p_0(x,t))}{p_0(x,t)^{1/2}} \cdot \frac{\partial_x p_0(x,t)}{p_0(x,t)^{1/2}} \, dx \right)^2$$

$$\leq \left( \int_{-\infty}^{\infty} \frac{(\partial_x p(x,t) - \partial_x p_0(x,t))^2}{p_0(x,t)} \, dx \right) \left( \int_{-\infty}^{\infty} \frac{(\partial_x p_0(x,t))^2}{p_0(x,t)} \, dx \right)$$

$$= Q_t' \int_{-\infty}^{\infty} s_0(x,t)^2 p_0(x,t) \, dx$$

$$\lesssim \frac{Q_t'}{t}.$$

Therefore, we have obtained a bound of order at most $\frac{Q_t'}{t}$. The proof is complete. $\qquad\square$

## E  LOW NOISE: DEFERRED PROOFS

This section contains the deferred proofs of Proposition 3 as well as Lemmas 5 and 6.

### E.1  PROOF OF PROPOSITION 3

*Proof of Proposition 3.* Let $C$ denote a sufficiently large universal constant, and define three regions

$$D_1 := \left\{ x \in \mathbb{R} : |x| < 1 - \sqrt{Ct\log(1/t)} \right\},$$

$$D_2 := \left\{ x \in \mathbb{R} : 1 - \sqrt{Ct\log(1/t)} \leq |x| \leq 1 + C\sqrt{t} \right\},$$

$$D_3 := \left\{ x \in \mathbb{R} : |x| > 1 + C\sqrt{t} \right\}.$$

Then $\mathbb{F}_t(f \,||\, f_0) = I_1 + I_2 + I_3$ where $I_j = \int_{x \in D_j} |s(x,t) - s_0(x,t)|^2 \, p_0(x,t) \, dx$ for $j = 1, 2, 3$. Each term will be bounded separately.

**Bounding $I_1$:** For $x \in D_1$, we have $p_0(x,t) \asymp 1$ by Lemma 7. Moreover, $\partial_x p_0(x,t) = f_0(-1)\varphi_t(x+1) - f(1)\varphi_t(x-1) + \int_{-1}^{1} f_0'(\mu)\varphi_t(x-\mu) \, d\mu$. Since $|x| - 1 > \sqrt{Ct\log(1/t)}$

and $||f_0'||_\infty \lesssim 1$, it is immediately clear $|\partial_x p_0(x,t)|^2 \lesssim 1$. Therefore,

$$I_1 = \int_{x \in D_1} \frac{|\partial_x p(x,t)p_0(x,t) - \partial_x p_0(x,t)p(x,t)|^2}{p_0(x,t)^2 p(x,t)^2} p_0(x,t)\, dx$$

$$\asymp \int_{x \in D_1} |\partial_x p(x,t)p_0(x,t) - \partial_x p_0(x,t)p(x,t)|^2\, dx$$

$$\lesssim \int_{x \in D_1} |p_0(x,t)|^2 |\partial_x p(x,t) - \partial_x p_0(x,t)|^2\, dx + \int_{x \in D_1} |\partial_x p_0(x,t)|^2 |p(x,t) - p_0(x,t)|^2\, dx$$

$$\lesssim \int_{-\infty}^\infty |\partial_x p(x,t) - \partial_x p_0(x,t)|^2\, dx + \int_{-\infty}^\infty |p(x,t) - p_0(x,t)|^2\, dx$$

$$\lesssim Q' + Q \tag{35}$$

where we have used $p(x,t) \asymp p_0(x,t)$. We have also used $||h * \varphi_t|| \lesssim ||h||$ for any function $h : \mathbb{R} \to \mathbb{R}$.

**Bounding $I_2$:** For $x \in D_2$, we still have $p_0(x,t) \asymp 1$. Then $I_2 \lesssim \int_{x \in D_2} |\partial_x p_0(x,t)p(x,t) - \partial_x p(x,t)p_0(x,t)|^2\, dx$. Consider

$$\partial_x p(x,t) = f(-1)\varphi_t(x+1) - f(1)\varphi_t(x-1) + \int_{-1}^1 f'(\mu)\varphi_t(x-\mu)\, d\mu,$$

$$\partial_x p_0(x,t) = f_0(-1)\varphi_t(x+1) - f_0(1)\varphi_t(x-1) + \int_{-1}^1 f_0'(\mu)\varphi_t(x-\mu)\, d\mu.$$

Therefore,

$$\partial_x p_0(x,t)p(x,t) = f_0(-1)\varphi_t(x+1)p(x,t) - f_0(1)\varphi_t(x-1)p(x,t) + p(x,t)\int_{-1}^1 f_0'(\mu)\varphi_t(x-\mu)\, d\mu,$$

$$\partial_x p(x,t)p_0(x,t) = f(-1)\varphi_t(x+1)p_0(x,t) - f_0(1)\varphi_t(x-1)p_0(x,t) + p_0(x,t)\int_{-1}^1 f'(\mu)\varphi_t(x-\mu)\, d\mu,$$

and so it follows

$$\int_{x \in D_2} |\partial_x p_0(x,t)p(x,t) - \partial_x p(x,t)p_0(x,t)|^2\, dx$$

$$\lesssim \int_{x \in D_2} |\varphi_t(x+1)|^2 |f_0(-1)p(x,t) - f(-1)p_0(x,t)|^2 + |\varphi_t(x-1)|^2 |f_0(1)p(x,t) - f(1)p_0(x,t)|^2\, dx$$

$$+ \int_{x \in D_2} \left( p(x,t)\int_{-1}^1 \varphi_t(x-\mu)f_0'(\mu)\, d\mu - p_0(x,t)\int_{-1}^1 \varphi_t(x-\mu)f'(\mu)\, d\mu \right)^2 dx. \tag{36}$$

Let us look at the second term in (36). Observe

$$\int_{x \in D_2} \left( p(x,t)\int_{-1}^1 \varphi_t(x-\mu)f_0'(\mu)\, d\mu - p_0(x,t)\int_{-1}^1 \varphi_t(x-\mu)f'(\mu)\, d\mu \right)^2 dx$$

$$\lesssim \int_{x \in D_2} \left( \int_{-1}^1 \varphi_t(x-\mu)f_0'(\mu)\, d\mu \right)^2 (p(x,t) - p_0(x,t))^2\, dx$$

$$+ \int_{x \in D_2} p_0(x,t)^2 \left( \int_{-1}^1 \varphi_t(x-\mu)(f_0'(\mu) - f'(\mu))\, d\mu \right)^2 dx$$

$$\lesssim \int_{x \in D_2} (p(x,t) - p_0(x,t))^2\, dx + \int_{x \in D_2} \left( \int_{-1}^1 \varphi_t(x-\mu)(f_0'(\mu) - f'(\mu))\, d\mu \right)^2 dx$$

$$\lesssim \int_{-\infty}^\infty |p(x,t) - p_0(x,t)|^2\, dx + \int_{-\infty}^\infty |((f-f_0)') * \varphi_t(x)|^2\, dx$$

$$\lesssim Q + Q'.$$

Here, we have used $\alpha \geq 1$ to assert $\left( \int_{-1}^{1} \varphi_t(x - \mu) f_0'(\mu) \, d\mu \right)^2 \lesssim 1$. Let us now look at the first term in (36). Without loss of generality, take nonnegative $x \in D_2$, namely $x \in D_2 \cap \mathbb{R}_+$. The argument for negative $x$ is entirely analogous. It is straightforward to see $|\varphi_t(x + 1)|^2 |f_0(-1)p(x, t) - f(-1)p_0(x, t)|^2 \lesssim \frac{1}{t} e^{-\frac{1}{t}}$. Looking at the remaining term, consider that $f(1) = f_0(1)$ since $f - f_0$ is $\alpha$-Hölder on $\mathbb{R}$ and $f, f_0$ are only supported on $[-1, 1]$. Thus, it follows

$$|f(1)p_0(x, t) - f_0(1)p(x, t)|^2$$

$$= \left| f(1) \int_{-1}^{1} \varphi_t(x - \mu) f_0(\mu) \, d\mu - f_0(1) \int_{-1}^{1} \varphi_t(x - \mu) f(\mu) \, d\mu \right|^2$$

$$= |f_0(1)|^2 \left| \int_{-1}^{1} \varphi_t(x - \mu)(f_0 - f)(\mu) \, d\mu \right|^2$$

$$= |f_0(1)|^2 \left| \sum_{k=0}^{\lfloor \alpha \rfloor - 1} \frac{(f_0 - f)^{(k)}(1)}{k!} \int_{-1}^{1} (\mu - 1)^k \varphi_t(x - \mu) \, d\mu + \frac{1}{\lfloor \alpha \rfloor!} \int_{-1}^{1} (f_0 - f)^{(\lfloor \alpha \rfloor)}(\xi)(\mu - 1)^{\lfloor \alpha \rfloor} \varphi_t(x - \mu) \, d\mu \right|^2$$

where $\xi$ is some point between $\mu$ and $1$. Furthermore, consider $(f_0 - f)^{(k)}(1) = 0$ for all $1 \leq k \leq \lfloor \alpha \rfloor$ since $f - f_0 \in \mathcal{H}_\alpha(\mathbb{R})$. Therefore,

$$|f_0(1)|^2 \left| \sum_{k=0}^{\lfloor \alpha \rfloor - 1} \frac{(f_0 - f)^{(k)}(1)}{k!} \int_{-1}^{1} (\mu - 1)^k \varphi_t(x - \mu) \, d\mu + \frac{1}{\lfloor \alpha \rfloor!} \int_{-1}^{1} (f_0 - f)^{(\lfloor \alpha \rfloor)}(\xi)(\mu - 1)^{\lfloor \alpha \rfloor} \varphi_t(x - \mu) \, d\mu \right|^2$$

$$= |f_0(1)|^2 \left| \frac{1}{\lfloor \alpha \rfloor!} \int_{-1}^{1} ((f_0 - f)^{(\lfloor \alpha \rfloor)}(\xi) - (f_0 - f)^{(\lfloor \alpha \rfloor)}(1))(\mu - 1)^{\lfloor \alpha \rfloor} \varphi_t(x - \mu) \, d\mu \right|^2$$

$$\lesssim \int_{-1}^{1} \left| ((f_0 - f)^{(\lfloor \alpha \rfloor)}(\xi) - (f_0 - f)^{(\lfloor \alpha \rfloor)}(1)) \right|^2 |\mu - 1|^{2\lfloor \alpha \rfloor} \varphi_t(x - \mu) \, d\mu$$

$$\lesssim \int_{-1}^{1} |\mu - 1|^{2(\alpha - \lfloor \alpha \rfloor)} |\mu - 1|^{2\lfloor \alpha \rfloor} \varphi_t(x - \mu) \, d\mu$$

$$\lesssim (t \log(1/t))^\alpha.$$

Here, we have used the Hölder property of $f - f_0$ and that nonnegative $x \in D_2$ implies $|x - 1| \leq \sqrt{Ct \log(1/t)}$. Therefore, we have shown

$$\int_{x \in D_2 \cap \mathbb{R}_+} |\varphi_t(x + 1)|^2 |f_0(-1)p(x, t) - f(-1)p_0(x, t)|^2 \, dx + |\varphi_t(x - 1)|^2 |f_0(1)p(x, t) - f(1)p_0(x, t)|^2 \, dx$$

$$\lesssim \int_{x \in D_2 \cap \mathbb{R}_+} \frac{1}{t} e^{-\frac{1}{t}} + (t \log(1/t))^\alpha \, dx$$

$$\lesssim \sqrt{\frac{\log(1/t)}{t}} e^{-\frac{1}{t}} + (t \log(1/t))^{\alpha - 1/2}$$

$$\lesssim t^{\alpha - 1}.$$

The argument for $x \in D_2 \cap \mathbb{R}_-$ is entirely analogous, so we have shown the bound $t^{\alpha - 1}$ for the first term in (36). To summarize, we have thus proved

$$I_2 \lesssim Q + Q' + t^{\alpha - 1}. \tag{37}$$

**Bounding $I_3$:** When $x \in D_3$, we no longer have $p_0(x,t) \asymp 1$. However, since $p(x,t) \asymp p_0(x,t)$, we have

$$
I_3 = \int_{x \in D_3} \frac{|\partial_x p(x,t) p_0(x,t) - \partial_x p_0(x,t) p(x,t)|^2}{p(x,t)^2 p_0(x,t)^2} p_0(x,t)\, dx
$$

$$
\asymp \int_{x \in D_3} \frac{|\partial_x p(x,t) p_0(x,t) - \partial_x p_0(x,t) p(x,t)|^2}{p_0(x,t)^3}\, dx
$$

$$
\lesssim \int_{x \in D_3} \frac{|\varphi_t(x+1)|^2}{p_0(x,t)^3} |f_0(-1)p(x,t) - f(-1)p_0(x,t)|^2 + \frac{|\varphi_t(x-1)|^2}{p_0(x,t)^3} |f_0(1)p(x,t) - f(1)p_0(x,t)|^2\, dx
\tag{38}
$$

$$
+ \int_{x \in D_3} \frac{1}{p_0(x,t)^3} \left( p(x,t) \int_{-1}^{1} \varphi_t(x-\mu) f_0'(\mu)\, d\mu - p_0(x,t) \int_{-1}^{1} \varphi_t(x-\mu) f'(\mu)\, d\mu \right)^2 dx
\tag{39}
$$

where we have followed the analysis of $I_2$ to bound the numerator. Let us look at (38) first. Take $x \in D_3$, and without loss of generality take $x$ to be nonnegative, i.e. $x \in D_3 \cap \mathbb{R}_+$. It is straightforward to see that $\frac{|\varphi_t(x+1)|^2}{p_0(x,t)^3} |f_0(-1)p(x,t) - f(-1)p_0(x,t)|^2 \lesssim \frac{|\varphi_t(x+1)|^2}{p_0(x,t)} \lesssim e^{-\frac{c}{t}} p_0(x,t)$ where $c > 0$ is a small universal constant. Therefore, $\int_{x \in D_3 \cap \mathbb{R}_+} \frac{|\varphi_t(x+1)|^2}{p_0(x,t)^3} |f_0(-1)p(x,t) - f(-1)p_0(x,t)|^2\, dx \lesssim e^{-c/t}$. Looking at the other term in (38), we can follow the same calculation in the analysis of $I_2$ to get

$$
|f(1)p_0(x,t) - f_0(1)p(x,t)|^2 \lesssim \left| \int_{-1}^{1} ((f_0 - f)^{(\lfloor \alpha \rfloor)}(\xi) - (f_0 - f)^{(\lfloor \alpha \rfloor)}(1))(\mu - 1)^{\lfloor \alpha \rfloor} \varphi_t(x-\mu)\, d\mu \right|^2
$$

$$
\lesssim \left| \int_{-1}^{1} |\mu - 1|^{\alpha} \varphi_t(x-\mu)\, d\mu \right|^2
$$

$$
\leq e^{-\frac{3(x-1)^2}{4t}} \left| \int_{-1}^{1} |\mu - 1|^{\alpha} \cdot \frac{1}{\sqrt{2\pi t}} e^{-\frac{(x-\mu)^2}{8t}}\, d\mu \right|^2
$$

$$
\lesssim t^{\alpha} e^{-\frac{3(x-1)^2}{4t}}.
$$

Therefore, by Lemma 7, we have

$$
\int_{x \in D_3 \cap \mathbb{R}_+} \frac{|\varphi_t(x-1)|^2}{p_0(x,t)^3} |f_0(1)p(x,t) - f(1)p_0(x,t)|^2\, dx
$$

$$
\lesssim t^{\alpha} \int_{x \in D_3 \cap \mathbb{R}_+} \frac{1}{2\pi t} e^{-\frac{(x-1)^2}{t}} \left( 1 \vee \frac{x-1}{\sqrt{t}} \right)^3 e^{\frac{3(x-1)^2}{2t}} e^{-\frac{3(x-1)^2}{4t}}\, dx
$$

$$
\lesssim t^{\alpha - 1/2} \int_{x \in D_3 \cap \mathbb{R}_+} \left( 1 \vee \frac{x-1}{\sqrt{t}} \right)^3 \frac{1}{\sqrt{2\pi t}} e^{-\frac{(x-1)^2}{4t}}\, dx
$$

$$
\lesssim t^{\alpha - 1/2}.
$$

Hence, we have shown (38) $\lesssim e^{-c/t} + t^{\alpha - 1/2} \lesssim t^{\alpha - 1}$.

It remains to bound (39). Observe

$$
\int_{x \in D_3} \frac{1}{p_0(x,t)^3} \left( p(x,t) \int_{-1}^{1} \varphi_t(x-\mu) f_0'(\mu)\, d\mu - p_0(x,t) \int_{-1}^{1} \varphi_t(x-\mu) f'(\mu)\, d\mu \right)^2 dx
$$

$$
\lesssim \int_{x \in D_3} \frac{\left( \int_{-1}^{1} \varphi_t(x-\mu) f_0'(\mu)\, d\mu \right)^2}{p_0(x,t)^3} (p(x,t) - p_0(x,t))^2\, dx
$$

$$
+ \int_{x \in D_3} \frac{1}{p_0(x,t)} \left( \int_{-1}^{1} \varphi_t(x-\mu)(f - f_0)'(\mu)\, d\mu \right)^2 dx
$$

$$
\lesssim \int_{x \in D_3} \frac{1}{p_0(x,t)} (p(x,t) - p_0(x,t))^2\, dx + \int_{x \in D_3} \frac{1}{p_0(x,t)} \left( \int_{-1}^{1} \varphi_t(x-\mu)(f - f_0)'(\mu)\, d\mu \right)^2 dx.
$$

Here, we have used $||f_0'||_\infty \lesssim 1$ since $\alpha \geq 1$. To proceed with the argument, let us write $g = f - f_0$ and note $p(x,t) - p_0(x,t) = (g * \varphi_t)(x)$. Observe we have

$$|(g * \varphi_t)(x)| = \left| \int_{-1}^1 g(\mu)\varphi_t(x-\mu)\, d\mu \right|$$

$$= \left| \sum_{k=0}^{\lfloor \alpha \rfloor - 1} \frac{g^{(k)}(1)}{k!} \int_{-1}^1 (\mu-1)^k \varphi_t(x-\mu)\, d\mu + \frac{1}{\lfloor \alpha \rfloor !} \int_{-1}^1 g^{(\lfloor \alpha \rfloor)}(\xi)(\mu-1)^{\lfloor \alpha \rfloor} \varphi_t(x-\mu)\, d\mu \right|$$

$$= \left| \frac{1}{\lfloor \alpha \rfloor !} \int_{-1}^1 (g^{(\lfloor \alpha \rfloor)}(\xi) - g^{\lfloor \alpha \rfloor}(1))(\mu-1)^{\lfloor \alpha \rfloor} \varphi_t(x-\mu)\, d\mu \right|$$

$$\lesssim \int_{-1}^1 |\mu - 1|^{\alpha - \lfloor \alpha \rfloor} \cdot |\mu - 1|^{\lfloor \alpha \rfloor} \varphi_t(x-\mu)\, d\mu$$

$$= \int_{-1}^1 |\mu - 1|^\alpha \varphi_t(x-\mu)\, d\mu.$$

Here, $\xi$ is some point between $\mu$ and 1. We have used $g^{(k)}(1) = 0$ for all $1 \leq k \leq \lfloor \alpha \rfloor$. For $x \in D_3$, we have $|x| > 1 + C\sqrt{t}$. Then we have

$$\int_{-1}^1 |\mu - 1|^\alpha \varphi_t(x-\mu)\, d\mu \leq e^{-\frac{3(|x|-1)^2}{8t}} \int_{-1}^1 |\mu - 1|^\alpha \frac{1}{\sqrt{2\pi t}} e^{-\frac{(x-\mu)^2}{8t}}\, d\mu \lesssim t^{\alpha/2} e^{-\frac{3(|x|-1)^2}{8t}}.$$

Therefore,

$$\int_{x \in D_3} \frac{1}{p_0(x,t)}(p(x,t) - p_0(x,t))^2\, dx \lesssim t^\alpha \int_{x \in D_3} \frac{e^{-\frac{3(|x|-1)^2}{4t}}}{p_0(x,t)}\, dx$$

$$\lesssim t^\alpha \int_{x \in D_3} \left( 1 \vee \frac{|x|-1}{\sqrt{t}} \right) e^{-\frac{(|x|-1)^2}{4t}}\, dx$$

$$\lesssim t^\alpha \int_{|x| > 1 + C\sqrt{t}} e^{-\frac{(|x|-1)^2}{8t}}\, dx$$

$$\asymp t^\alpha.$$

A similar argument shows

$$\int_{x \in D_3} \frac{1}{p_0(x,t)} \left( \int_{-1}^1 \varphi_t(x-\mu)(f-f_0)'(\mu)\, d\mu \right)^2\, dx \lesssim t^{\alpha-1}.$$

Therefore, we have shown $(39) \lesssim t^\alpha + t^{\alpha-1} \lesssim t^{\alpha-1}$. Putting together our bounds, we have shown $I_3 \lesssim t^{\alpha-1}$, which completes the proof. $\qquad\square$

### E.2 PROOFS OF LEMMAS 5 AND 6

*Proof of Lemma 5.* Since $E\left( |\hat{Q}_K - Q|^2 \right) = |E(\hat{Q}_K) - Q|^2 + \text{Var}(\hat{Q}_K)$, it suffices to bound the bias and variance separately. Writing the expansion $f = \sum_{k=1}^\infty \theta_k \psi_k$ and noting $Q = \sum_{k=1}^\infty (\theta_k - \theta_{0,k})^2$, observe $E(\hat{Q}_K) = \sum_{k=1}^K (\theta_k - \theta_{0,k})^2$ and so the squared bias satisfies

$$|E(\hat{Q}_K) - Q|^2 \leq \left| \sum_{k=K+1}^\infty (\theta_k - \theta_{0,k})^2 \right|^2 = \left| \sum_{k=K+1}^\infty k^{-2\alpha} \cdot k^{2\alpha}(\theta_k - \theta_{0,k})^2 \right|^2 \lesssim K^{-4\alpha}$$

since the Fourier coefficients of $f - f_0$ satisfy $\sum_{k=1}^\infty k^{2\alpha}(\theta_k - \theta_{0,k})^2 \lesssim 1$. Moving to the variance, consider

$$\text{Var}(\hat{Q}_K) = \frac{1}{\binom{n}{2}} \sum_{i \neq j} \sum_{r \neq s} \text{Cov}\left( \sum_{k=1}^K (\psi_k(\mu_i) - \theta_{0,k})(\psi_k(\mu_j) - \theta_{0,k}), \sum_{k=1}^K (\psi_k(\mu_r) - \theta_{0,k})(\psi_k(\mu_s) - \theta_{0,k}) \right).$$

If $\{i, j\} \cap \{r, s\} = \emptyset$, the covariance vanishes due to independence. There are $O(n^2)$ choices of indices such that $i \neq j, r \neq s$, and $|\{i, j\} \cap \{r, s\}| = 2$. Similarly, there are $O(n^3)$ choices for which $i \neq j, r \neq s$, and $|\{i, j\} \cap \{r, s\}| = 1$. Let $N_1$ and $N_2$ denote the respective counts of choices. Hence, by the identical distribution of the $\mu_i$'s, it follows

$$
\mathrm{Var}\left(\hat{Q}_K\right) = \frac{N_1}{\binom{n}{2}^2} \mathrm{Var}\left(\sum_{k=1}^{K} (\psi_k(\mu_1) - \theta_{0,k})(\psi_k(\mu_2) - \theta_{0,k})\right)
$$

$$
+ \frac{N_2}{\binom{n}{2}^2} \mathrm{Cov}\left(\sum_{k=1}^{K} (\psi_k(\mu_1) - \theta_{0,k})(\psi_k(\mu_2) - \theta_{0,k}), \sum_{k=1}^{K} (\psi_k(\mu_1) - \theta_{0,k})(\psi_k(\mu_3) - \theta_{0,k})\right).
$$

$$(40)$$

Each term of (40) will be bounded separately.

**First term in (40):** By direct calculation, we have

$$
\mathrm{Var}\left(\sum_{k=1}^{K} (\psi_k(\mu_1) - \theta_{0,k})(\psi_k(\mu_2) - \theta_{0,k})\right)
$$

$$
= \sum_{k=1}^{K} \sum_{k'=1}^{K} \mathrm{Cov}((\psi_k(\mu_1) - \theta_{0,k})(\psi_k(\mu_2) - \theta_{0,k}), (\psi_{k'}(\mu_1) - \theta_{0,k'})(\psi_{k'}(\mu_2) - \theta_{0,k'}))
$$

$$
= \sum_{k=1}^{K} \sum_{k'=1}^{K} \left(E\left((\psi_k(\mu_1) - \theta_{0,k})(\psi_{k'}(\mu_1) - \theta_{0,k'}))\right)\right)^2 - (\theta_k - \theta_{0,k})^2(\theta_{k'} - \theta_{0,k'})^2
$$

$$
\leq \sum_{k=1}^{K} \sum_{k'=1}^{K} \left(\int_{-1}^{1} (\psi_k(\mu) - \theta_{0,k})(\psi_{k'}(\mu) - \theta_{0,k'}) f(\mu) \, d\mu\right)^2
$$

$$
= \sum_{k=1}^{K} \sum_{k'=1}^{K} \int_{-1}^{1} \int_{-1}^{1} (\psi_k(\mu) - \theta_{0,k})(\psi_{k'}(\mu) - \theta_{0,k'})(\psi_k(\nu) - \theta_{0,k})(\psi_{k'}(\nu) - \theta_{0,k'}) f(\mu) f(\nu) \, d\mu \, d\nu
$$

$$
= \int_{-1}^{1} \int_{-1}^{1} \left(\sum_{k=1}^{K} (\psi_k(\mu) - \theta_{0,k})(\psi_k(\nu) - \theta_{0,k})\right)^2 f(\mu) f(\nu) \, d\mu \, d\nu
$$

$$
\lesssim \int_{-1}^{1} \int_{-1}^{1} \left(\sum_{k=1}^{K} (\psi_k(\mu) - \theta_{0,k})(\psi_k(\nu) - \theta_{0,k})\right)^2 d\mu \, d\nu
$$

$$
= \sum_{k=1}^{K} \sum_{k'=1}^{K} \int_{-1}^{1} \int_{-1}^{1} (\psi_k(\mu) - \theta_{0,k})(\psi_{k'}(\mu) - \theta_{0,k'})(\psi_k(\nu) - \theta_{0,k})(\psi_{k'}(\nu) - \theta_{0,k'}) \, d\mu \, d\nu
$$

$$
= \sum_{k=1}^{K} \sum_{k'=1}^{K} \left(\int_{-1}^{1} (\psi_k(\mu) - \theta_{0,k})(\psi_{k'}(\mu) - \theta_{0,k'}) \, d\mu\right)^2
$$

$$
= \sum_{k=1}^{K} \sum_{k'=1}^{K} \left(\int_{-1}^{1} \psi_k(\mu)\psi_{k'}(\mu) \, d\mu - \theta_{0,k'} \int_{-1}^{1} \psi_k(\mu) \, d\mu - \theta_{0,k} \int_{-1}^{1} \psi_{k'}(\mu) \, d\mu + 2\theta_{0,k}\theta_{0,k'}\right)^2
$$

$$
\lesssim \sum_{k=1}^{K} \sum_{k'=1}^{K} \left(\int_{-1}^{1} \psi_k(\mu)\psi_{k'}(\mu) \, d\mu\right)^2 + \theta_{0,k'}^2 \left(\int_{-1}^{1} \psi_k(\mu) \, d\mu\right)^2 + \theta_{0,k}^2 \left(\int_{-1}^{1} \psi_{k'}(\mu) \, d\mu\right)^2 + 4\theta_{0,k}^2\theta_{0,k'}^2
$$

$$
\lesssim K + \sum_{k=1}^{K} \theta_{0,k}^2 + \left(\sum_{k=1}^{K} \theta_{0,k}^2\right)^2
$$

$$
\lesssim K + \sum_{k=1}^{K} k^{2\alpha}\theta_{0,k}^2 + \left(\sum_{k=1}^{K} k^{2\alpha}\theta_{0,k}^2\right)^2
$$

$$
\lesssim K.
$$

Here, we have used $\int_{-1}^{1} \psi_k^2(\mu) \, d\mu = 1$, $\psi_1 \equiv \frac{1}{\sqrt{2}}$, and $\{\psi_k\}_{k=1}^{\infty}$ is an orthonormal basis to conclude $\int_{-1}^{1} \psi_k(\mu) \, d\mu = 0$ for $k \geq 2$. The final line follows from $\sum_{k=1}^{\infty} \theta_{0,k}^2 k^{2\alpha} \lesssim 1$. Therefore, the first term in (40) is of order at most $\mathrm{Var}(\hat{Q}_K) \lesssim \frac{K}{n^2}$.

**Second term in (40):** Again, direct calculation gives

$$\mathrm{Cov}\left( \sum_{k=1}^{K} (\psi_k(\mu_1) - \theta_{0,k})(\psi_k(\mu_2) - \theta_{0,k}), \sum_{k=1}^{K} (\psi_k(\mu_1) - \theta_{0,k})(\psi_k(\mu_3) - \theta_{0,k}) \right)$$

$$= \sum_{k=1}^{K} \sum_{k'=1}^{K} \mathrm{Cov}((\psi_k(\mu_1) - \theta_{0,k})(\psi_k(\mu_2) - \theta_{0,k}), (\psi_{k'}(\mu_1) - \theta_{0,k'})(\psi_{k'}(\mu_3) - \theta_{0,k'}))$$

$$= \sum_{k=1}^{K} \sum_{k'=1}^{K} E\left( (\psi_k(\mu_1) - \theta_{0,k})(\psi_{k'}(\mu_1) - \theta_{0,k'}) \right) (\theta_k - \theta_{0,k})(\theta_{k'} - \theta_{0,k'}) - (\theta_k - \theta_{0,k})^2 (\theta_{k'} - \theta_{0,k'})^2$$

$$\leq \sum_{k=1}^{K} \sum_{k'=1}^{K} \left( \int_{-1}^{1} (\psi_k(\mu) - \theta_{0,k})(\psi_{k'}(\mu) - \theta_{0,k'}) f(\mu) \, d\mu \right) (\theta_k - \theta_{0,k})(\theta_{k'} - \theta_{0,k'})$$

$$= \int_{-1}^{1} \left( \sum_{k=1}^{K} (\psi_k(\mu) - \theta_{0,k})(\theta_k - \theta_{0,k}) \right)^2 f(\mu) \, d\mu$$

$$\lesssim \int_{-1}^{1} \left( \sum_{k=1}^{K} (\psi_k(\mu) - \theta_{0,k})(\theta_k - \theta_{0,k}) \right)^2 d\mu$$

$$= \sum_{k=1}^{K} \sum_{k'=1}^{K} \left( \int_{-1}^{1} (\psi_k(\mu) - \theta_{0,k})(\psi_{k'}(\mu) - \theta_{0,k'}) \, d\mu \right) (\theta_k - \theta_{0,k})(\theta_{k'} - \theta_{0,k'})$$

$$= \sum_{k=1}^{K} \sum_{k'=1}^{K} \left( \mathbb{1}_{\{k=k'\}} - \theta_{0,k} \int_{-1}^{1} \psi_{k'}(\mu) \, d\mu - \theta_{0,k'} \int_{-1}^{1} \psi_k(\mu) \, d\mu + \theta_{0,k}\theta_{0,k'} \right) (\theta_k - \theta_{0,k})(\theta_{k'} - \theta_{0,k'})$$

$$= (\theta_k - \theta_{0,k})^2 + 2\left( \int_{-1}^{1} \psi_1(\mu) \, d\mu \right)(\theta_1 - \theta_{0,1}) \sum_{k=1}^{K} \theta_{0,k}(\theta_k - \theta_{0,k}) + \sum_{k=1}^{K} \sum_{k'=1}^{K} \theta_{0,k}\theta_{0,k'}(\theta_k - \theta_{0,k})(\theta_{k'} - \theta_{0,k'})$$

$$\lesssim \sum_{k=1}^{K} (\theta_k - \theta_{0,k})^2 + 0 + \left( \sum_{k=1}^{K} \theta_{0,k}(\theta_k - \theta_{0,k}) \right)^2$$

$$\lesssim \sum_{k=1}^{K} (\theta_k - \theta_{0,k})^2 + \left( \sum_{k=1}^{K} \theta_{0,k}^2 \right) \left( \sum_{k=1}^{K} (\theta_k - \theta_{0,k})^2 \right)$$

$$\lesssim Q.$$

Here, we have used $\int_{-1}^{1} \psi_k^2(\mu) \, d\mu = 1$, $\psi_1 \equiv \frac{1}{\sqrt{2}}$, and $\{\psi_k\}_{k=1}^{\infty}$ is an orthonormal basis to conclude $\int_{-1}^{1} \psi_k(\mu) \, d\mu = 0$ for $k \geq 2$. We have also used that $\theta_1 - \theta_{0,1} = \frac{1}{\sqrt{2}} \int f - f_0 = 0$ since both $f$ and $f_0$ integrate to one. We have also used Cauchy-Schwarz to obtain the penultimate line. Finally, we used $\sum_{k=1}^{K} \theta_{0,k}^2 \lesssim 1$ which is obtained by noting $\sum_{k=1}^{\infty} \theta_{0,k}^2 k^{2\alpha} \lesssim 1$. Hence, we have shown the second term in (40) is of order at most $\frac{Q}{n}$. The proof is complete. $\qquad\square$

*Proof of Lemma 6.* By the bias-variance decomposition, it suffices to bound the bias and variance separately. Note $Q' = \sum_{k=2}^{\infty} (\tilde{\theta}_k - \tilde{\theta}_{0,k})^2$. The squared bias satisfies

$$|E(\hat{Q}_K') - Q'|^2$$

$$= \left| \sum_{k=K+1}^{\infty} (\tilde{\theta}_k - \tilde{\theta}_{0,k})^2 \right|^2 = \left| \sum_{k=K+1}^{\infty} k^2 (\theta_k - \theta_{0,k})^2 \right|^2 \lesssim K^{-2(\alpha-1)} \left| \sum_{k=K+1}^{\infty} k^{2\alpha} (\theta_k - \theta_{0,k})^2 \right|^2 \lesssim K^{-2(\alpha-1)}.$$

Here, we have used the property that the Fourier coefficients of $f - f_0$ live in $\Theta_\alpha(L)$ defined in Section 3.1, and so $\sum_{k=1}^\infty k^{2\alpha}(\theta_k - \theta_{0,k})^2 \lesssim 1$.

Moving to the variance, consider

$$\text{Var}(\hat{Q}_K) = \frac{1}{\binom{n}{2}} \sum_{i \neq j} \sum_{r \neq s} \text{Cov}\left(\sum_{k=1}^K A_k(\mu_i)A_k(\mu_j), \sum_{k=1}^K A_k(\mu_r)A_k(\mu_s)\right).$$

If $\{i,j\} \cap \{r,s\} = \emptyset$, the covariance vanishes due to independence. There are $O(n^2)$ choices of indices such that $i \neq j, r \neq s$, and $|\{i,j\} \cap \{r,s\}| = 2$. Similarly, there are $O(n^3)$ choices for which $i \neq j, r \neq s$, and $|\{i,j\} \cap \{r,s\}| = 1$. Let $N_1$ and $N_2$ denote the respective counts of choices. Hence, by the identical distribution of the $\mu_i$'s, it follows

$$\text{Var}\left(\hat{Q}_K\right) = \frac{N_1}{\binom{n}{2}^2} \text{Var}\left(\sum_{k=1}^K A_k(\mu_1)A_k(\mu_2)\right) + \frac{N_2}{\binom{n}{2}^2} \text{Cov}\left(\sum_{k=1}^K A_k(\mu_1)A_k(\mu_2), \sum_{k=1}^K A_k(\mu_1)A_k(\mu_3)\right).$$

$$(41)$$

Each term of (41) will be bounded separately.

**First term in (41):** By direct calculation, we have

$$\text{Var}\left(\sum_{k=1}^K A_k(\mu_1)A_k(\mu_2)\right)$$

$$= \sum_{k=1}^K \sum_{k'=1}^K \text{Cov}(A_k(\mu_1)A_k(\mu_2), A_{k'}(\mu_1)A_{k'}(\mu_2))$$

$$= \sum_{k=1}^K \sum_{k'=1}^K \left(E\left(A_k(\mu_1)A_{k'}(\mu_1)\right)\right)^2 - (E(A_k(\mu_1)))^2 (E(A_{k'}(\mu_1)))^2$$

$$\leq \sum_{k=1}^K \sum_{k'=1}^K \left(\int_{-1}^1 A_k(\mu)A_{k'}(\mu)f(\mu)\,d\mu\right)^2$$

$$= \sum_{k=1}^K \sum_{k'=1}^K \int_{-1}^1 \int_{-1}^1 A_k(\mu)A_{k'}(\mu)A_k(\nu)A_{k'}(\nu)f(\mu)f(\nu)\,d\mu\,d\nu$$

$$= \int_{-1}^1 \int_{-1}^1 \left(\sum_{k=1}^K A_k(\mu)A_k(\nu)\right)^2 f(\mu)f(\nu)\,d\mu\,d\nu$$

$$\lesssim \int_{-1}^1 \int_{-1}^1 \left(\sum_{k=1}^K A_k(\mu)A_k(\nu)\right)^2 d\mu\,d\nu$$

$$= \sum_{k=1}^K \sum_{k'=1}^K \int_{-1}^1 \int_{-1}^1 A_k(\mu)A_{k'}(\mu)A_k(\nu)A_{k'}(\nu)\,d\mu\,d\nu$$

$$= \sum_{k=1}^K \sum_{k'=1}^K \left(\int_{-1}^1 A_k(\mu)A_{k'}(\mu)\,d\mu\right)^2.$$

Consider

$$\left(\int_{-1}^1 A_k(\mu)A_{k'}(\mu)\,d\mu\right)^2$$

$$= \left(\int_{-1}^1 \left((\pi k \psi_{2k+1}(\mu) - \tilde{\theta}_{0,2k}) + (-\pi k \psi_{2k}(\mu) - \tilde{\theta}_{0,2k+1})\right)\right.$$

$$\left. \left((\pi k' \psi_{2k'+1}(\mu) - \tilde{\theta}_{0,2k'}) + (-\pi k' \psi_{2k'}(\mu) - \tilde{\theta}_{0,2k'+1})\right) d\mu\right)^2.$$

Therefore,

$$\sum_{k=1}^{K}\sum_{k'=1}^{K}\left(\int_{-1}^{1} A_k(\mu)A_{k'}(\mu)\,d\mu\right)^2 \lesssim \sum_{k=1}^{K}\sum_{k'=1}^{K}\left(\int_{-1}^{1}(\pi k\psi_{2k+1}(\mu)-\tilde{\theta}_{0,2k})(\pi k'\psi_{2k'+1}(\mu)-\tilde{\theta}_{0,2k'})\,d\mu\right)^2$$

$$+\left(\int_{-1}^{1}(-\pi k\psi_{2k}(\mu)-\tilde{\theta}_{0,2k+1})(-\pi k'\psi_{2k'}(\mu)-\tilde{\theta}_{0,2k'+1})\,d\mu\right)^2$$

By a similar calculation as that in the proof of Lemma 6, it follows

$$\sum_{k=1}^{K}\sum_{k'=1}^{K}\left(\int_{-1}^{1}(\pi k\psi_{2k+1}(\mu)-\tilde{\theta}_{0,2k})(\pi k'\psi_{2k'+1}(\mu)-\tilde{\theta}_{0,2k'})\,d\mu\right)^2$$

$$+\left(\int_{-1}^{1}(-\pi k\psi_{2k}(\mu)-\tilde{\theta}_{0,2k+1})(-\pi k'\psi_{2k'}(\mu)-\tilde{\theta}_{0,2k'+1})\,d\mu\right)^2$$

$$\lesssim \sum_{k=1}^{K}\sum_{k'=1}^{K}\left(kk'\int_{-1}^{1}\psi_k(\mu)\psi_{k'}(\mu)\,d\mu\right)^2 + \sum_{k=1}^{K}k^2\tilde{\theta}_{0,k}^2 + \left(\sum_{k=1}^{K}\tilde{\theta}_{0,k}^2\right)^2$$

$$=\sum_{k=1}^{K}k^4 + K^2\sum_{k=1}^{K}k^{2(\alpha-1)}\tilde{\theta}_{0,2k}^2 + \left(\sum_{k=1}^{K}k^{2(\alpha-1)}\tilde{\theta}_{0,k}^2\right)^2$$

$$\lesssim K^5 + K^2 + 1$$

$$\asymp K^5$$

Therefore, the first term in (40) is of order at most $\mathrm{Var}(\hat{Q}_K)\lesssim\frac{K^5}{n^2}$.

**Second term in (41):** A very similar calculation to that found in the proof of Lemma 5 (namely the bound for the second term in (40)) with the obvious modifications will show that the second term in (41) is of order at most $\frac{Q'}{n}$. We omit the details for brevity.

For notational ease, let us write $\rho_k = E(A_k(\mu_1))$. Direct calculation gives

$$\mathrm{Cov}\left(\sum_{k=1}^{K} A_k(\mu_1)A_k(\mu_2), \sum_{k=1}^{K} A_k(\mu_1)A_k(\mu_3)\right) = \sum_{k=1}^{K}\sum_{k'=1}^{K}\mathrm{Cov}(A_k(\mu_1)A_k(\mu_2), A_{k'}(\mu_1)A_{k'}(\mu_3))$$

$$=\sum_{k=1}^{K}\sum_{k'=1}^{K} E\left(A_k(\mu_1)A_{k'}(\mu_1)\right)\rho_k\rho_{k'} - \rho_k^2\rho_{k'}^2$$

$$\leq \sum_{k=1}^{K}\sum_{k'=1}^{K}\left(\int_{-1}^{1} A_k(\mu)A_{k'}(\mu)f(\mu)\,d\mu\right)\rho_k\rho_{k'}$$

$$=\int_{-1}^{1}\left(\sum_{k=1}^{K} A_k(\mu)\rho_k\right)^2 f(\mu)\,d\mu$$

$$\lesssim \int_{-1}^{1}\left(\sum_{k=1}^{K} A_k(\mu)\rho_k\right)^2 d\mu$$

$$=\sum_{k=1}^{K}\sum_{k'=1}^{K}\left(\int_{-1}^{1} A_k(\mu)A_{k'}(\mu)\,d\mu\right)\rho_k\rho_{k'}.$$

Following our earlier calculation bounding the first term in (40), we have

$$\sum_{k=1}^{K} \sum_{k'=1}^{K} \left( \int_{-1}^{1} A_k(\mu) A_{k'}(\mu) \, d\mu \right) \rho_k \rho_{k'}$$

$$\lesssim \sum_{k=1}^{K} \sum_{k'=1}^{K} \left| \int_{-1}^{1} A_k(\mu) A_{k'}(\mu) \, d\mu \right| |\rho_k \rho_{k'}|$$

$$\lesssim \sum_{k=1}^{2K+1} \sum_{k'=1}^{2K+1} \left( kk' \mathbb{1}_{\{k=k'\}} + k'|\tilde{\theta}_{0,k}| \left| \int_{-1}^{1} \psi_{k'}(\mu) \, d\mu \right| + k|\tilde{\theta}_{0,k'}| \left| \int_{-1}^{1} \psi_k(\mu) \, d\mu \right| + |\theta_{0,k} \theta_{0,k'}| \right) |\tilde{\theta}_k - \tilde{\theta}_{0,k}||\tilde{\theta}_{k'} - \tilde{\theta}_{0,k'}|$$

$$\lesssim \sum_{k=1}^{2K+1} k^2 |\tilde{\theta}_k - \tilde{\theta}_{0,k}|^2 + 0 + \sum_{k=1}^{2K+1} \sum_{k'=1}^{2K+1} |\tilde{\theta}_{0,k} \tilde{\theta}_{0,k'}||\tilde{\theta}_k - \tilde{\theta}_{0,k}||\tilde{\theta}_{k'} - \tilde{\theta}_{0,k'}|$$

$$\lesssim K^2 \sum_{k=1}^{2K+1} (\tilde{\theta}_k - \tilde{\theta}_{0,k})^2 + \left( \sum_{k=1}^{2K+1} \tilde{\theta}_{0,k}^2 \right) \left( \sum_{k=1}^{2K+1} (\tilde{\theta}_k - \tilde{\theta}_{0,k})^2 \right)$$

$$\lesssim K^2 Q'.$$

Hence, we have shown the second term in (41) is of order at most $\frac{K^2 Q'}{n}$. The proof is complete. $\qquad\square$

## F    AUXILIARY TOOLS

**Lemma 7.** *If $t < 1$, then*

$$p(x,t) \asymp \begin{cases} 1 & \text{if } |x| \leq 1, \\ \left(1 \wedge \frac{\sqrt{t}}{|x|-1}\right) e^{-\frac{(|x|-1)^2}{2t}} & \text{if } |x| > 1. \end{cases}$$

*Proof of Lemma 7.* Consider that we have $p(x,t) = \int_{-1}^{1} f(\mu)\varphi_t(x-\mu) \, d\mu \asymp \int_{-1}^{1} \varphi_t(x-\mu) \, d\mu = \int_{-1}^{1} \frac{1}{\sqrt{2\pi t}} e^{-\frac{|x-\mu|^2}{2t}} \, d\mu$. Here, we have used $c_d \leq f \leq C_d$ on its support. Consider

$$\int_{-1}^{1} \frac{1}{\sqrt{2\pi t}} e^{-\frac{|x-\mu|^2}{2t}} \, d\mu = P\{|N(x,t)| \leq 1\}.$$

If $|x| \leq 1$, then clearly $P\{|N(x,t)| \leq 1\} \asymp 1$, and so the claim is proved for this case. If $|x| > 1$, consider we can use Lemma 13[2] from (Dou et al., 2024). $\qquad\square$

**Lemma 8** (Lemma 14 (Dou et al., 2024)). *If $f \in \mathcal{F}_\alpha$ and $p(x,t) = (f * \varphi_t)(x)$, then the score function $s(x,t) = \partial_x \log p(x,t)$ satisfies*

$$|s(x,t)|^2 \leq \frac{2}{t} \log\left(\frac{1}{(2\pi t)^{1/2} \cdot p(x,t)}\right)$$

*for all $x \in \mathbb{R}$ and $t > 0$.*

---

[2] Note the statement of Lemma 13 has a typo, but its proof yields the expression stated in Lemma 7.

