# OpenReview forum: "Diffusion models are optimal for hypothesis testing"
_ICLR.cc/2026/Conference — ICLR 2026 Conference Withdrawn Submission_

### Official Review · Reviewer_FzhB · 2025-10-29

**Soundness:** 3
**Presentation:** 2
**Contribution:** 2
**Rating:** 4
**Confidence:** 1

**Summary:**

This article rigorously analyzed the statistical limits of diffusion models.

**Strengths:**

- Rigorous analysis of the diffusion models in hypothesis testing.

**Weaknesses:**

- No conclusion section.
- No numerical experiments to verify the theoretical findings.

**Questions:**

- Add the conclusion section.
- Add some numerical experiments for better understanding of this study.

---

### Official Review · Reviewer_yTtc · 2025-10-31

**Soundness:** 2
**Presentation:** 2
**Contribution:** 2
**Rating:** 2
**Confidence:** 2

**Summary:**

In principle, diffusion models could be used for hypothesis testing: one could ask whether two distributions are the same or different by not just comparing the distributions, but by comparing them with different levels of added noise. If one does this, how well does one do? Is this a good strategy or a bad strategy? The authors claim that it is a good strategy, and estimate how well one can do given different levels of noise. The bulk of the paper is math that supports their claim about how well one can do.

**Strengths:**

There is a lot of math, and I mostly don't doubt that it is correct. Lengthy appendices justify the authors' core claims (although the length of the appendices seems to mostly be because of many multi-line equations).

**Weaknesses:**

My issues with this paper are mostly related to its conceptual basis and presentation, although there are some potential technical issues.

**Conceptual basis.** My first impressions of the topic of this paper were quite different than my later impressions of it. For starters, I assumed "diffusion models" were involved in the sense that, there was some sort of model trained via denoising score-matching, and there was some sort of sampling process that looked like a PF-ODE or reverse process. But this doesn't seem to be true. The paper isn't really about "diffusion models" at all, but about comparing distributions given different levels of noise corruption. This is a statistical problem that isn't that related to training or sampling from diffusion models, beyond the fact that scores and noise appear.

Also, I was confused by the "hypothesis testing" framing. Part of my issue is that the motivation presented in the introduction (e.g., using diffusion models in hypothesis testing) is very different from what was actually done. I don't think "applying diffusion models for hypothesis testing" (line 71) is a good description of the work. Moreover, it's not clear to me why the considered hypothesis testing problem is interesting. (The authors cite old statistics work around line 170, but I don't think it relates much to diffusion models). I think the authors should substantially reframe their work and contributions.

**Presentation.** The structure of the paper is a bit odd. After some introduction (which, again, only weakly relates prior work to the problem actually considered) and a high-level explanation of the main results, about half the paper is a dense collection of theorem statements with little prose or intuition surrounding them. This means that not only is the core problem strangely motivated, but that it is not that easy to follow the main results. There are no figures or examples, and no discussion or conclusion. All of these things would make the paper much more readable.

**Potential technical issues.** I didn't follow the math in detail, but a few assumptions seem potentially problematic. The most important, which is briefly addressed in a footnote on page 2, is that the authors assume a one-dimensional problem throughout. They claim that nothing changes except more cumbersome notation in the higher-dimensional case, but I don't buy this. Higher-dimensional statistics can be quite different from one-dimensional statistics, and I wonder whether their arguments really hold in the higher-D case. Either the authors should sketch analogue proofs for the higher-D case, or not state that the higher-D case is probably the same. Even if the arguments essentially still hold, the minimax scalings may change in higher dimensions.

A more minor issue is that I'm not sure how valid some of the Fourier math is. Approximating bounded functions on a finite domain in a Fourier fashion requires that issues at the boundaries are carefully controlled.

Finally, a nitpick: the minimax testing rate (Eq. 11) is a little hard to parse. I didn't immediately recognize that the authors meant that they have different scalings in three regimes. It would be helpful to clarify this better in the text.

**Questions:**

1. How does this work relate to trained diffusion models?

2. Why should one expect the arguments to also hold in the higher-D case? Are the minimax scalings really expected to be the same?

3. Can the authors comment on boundary issues with Fourier approximations?

---

### Official Review · Reviewer_tXzF · 2025-11-01

**Soundness:** 3
**Presentation:** 3
**Contribution:** 3
**Rating:** 6
**Confidence:** 3

**Summary:**

This paper investigates the statistical optimality of diffusion models for hypothesis testing problems. The authors study two main testing scenarios: (1) testing in Fisher divergence at each noise level t, formulated as detecting whether $F_t(f || f_0) = 0$ vs  $F_t(f || f_0)\ge {\epsilon}^2_t$, and (2) testing in total variation distance. The paper derives minimax testing rates for both problems and shows that appropriately designed tests based on diffusion's score-based paradigm achieve these optimal rates.

**Strengths:**

The paper addresses a fundamental gap in understanding diffusion models' capabilities beyond generation, providing rigorous statistical theory for hypothesis testing. Also solves a fundamental gap in understanding diffusion models' capabilities beyond generation which provides rigorous statistical theory for hypothesis testing. The counterintuitive result that testing becomes easier as noise increases (for t <- 1) due to increased smoothness is interesting. And technical part appears solid with detailed proofs.

**Weaknesses:**

fisrt, the paper focuses exclusively on statistical optimality without addressing computational aspects. The tests constructed (particularly the aggregated test) may be computationally prohibitive. Given that diffusion models' appeal partly lies in their practical success, this is a significant limitation. Also, the assumptions look a little bit restrictive: assumption 1 (periodicity of f_0) is stated to be wlg via inverse CDF transform, but this transformation affects the smoothness class and may not preserve the Holder property in general. The lower bound constructions largely follow Dou et al. (2024), and the upper bounds use classical orthogonal series estimation (Laurent 1996). The main novelty seems to be in assembling these pieces for the diffusion testing context. Also might consider the connection with actual diffusion model implementations.

**Questions:**

The paper restricts to d=1 "to maintain focus on mathematical essence." While understandable, at least discussing the challenges and expected results in higher dimensions would be valuable? Real implementations use $\hat{s}$ with estimation error. How much estimation error can be tolerated before the optimality guarantees break down? The alternative $f \in F$ is very broad (only boundedness constraints). Are there other natural alternative classes where diffusion testing might have particular advantages?

---

### Official Review · Reviewer_VUif · 2025-11-02

**Soundness:** 3
**Presentation:** 2
**Contribution:** 2
**Rating:** 2
**Confidence:** 2

**Summary:**

The paper studies the theoretical properties of diffusion models for hypothesis testing, not generation. It formulates testing via Fisher divergence at a noise level t and asks for the minimax separation rate of this test. It then aggregates tests across t to obtain a procedure that achieves the minimax rate for total variation (TV) testing under Hölder smoothness.  Overall, the contribution is a theoretical characterization of statistical limits of diffusion-based testing.

**Strengths:**

It studies the properties of diffusion model from a new perspective, framing diffusion as testing equality of noised score fields via Fisher divergence, with a precise minimax question.

It derives noise–level–dependent minimax rates in Fisher divergence and shows that aggregating over noise yields minimax-optimal TV testing under Hölder smoothness.

**Weaknesses:**

Theoretical, little real-world implication as written. The paper itself positions the contribution as characterizing statistical limits. The work provides rates and testing limits but no implemented procedure or empirical connection with in real-world problems.  It may fit better for mathematical venue.


Practical guidance is absent: Thresholds, finite-sample constants, and computational aspects (e.g., estimating scores/test statistics in practice) are not addressed, limiting applicability.

**Questions:**

This paper studies the hypothesis-testing properties of diffusion models and is squarely a theoretical statistics/learning theory contribution. It does not presently yield direct real-world implications or plug-and-play algorithms, so it feels better suited to mathematical statistics or learning theory venues than an ML conference.

---

### Note · Authors · 2025-11-12

I have read and agree with the venue's withdrawal policy on behalf of myself and my co-authors.